

# Global Patterns and Trends in Ground-Level Ozone Chemical Formation Regimes from 1996 to 2022

Yu Tian[1], Siyi Wang[1], Xiaomeng Jin[1, *]

[1]Department of Environmental Sciences, Rutgers, The State University of New Jersey, New Brunswick, NJ 08901, U.S.A

*Correspondence to*: Xiaomeng Jin (xiaomeng.jin@rutgers.edu)

**Abstract.** Ground-level ozone ($O_3$) formation in urban areas is nonlinearly dependent on the relatively availability of its precursors: oxides of nitrogen ($NO_x$) and volatile organic compounds (VOCs). To mitigate $O_3$ pollution, a crucial question is to identify the $O_3$ formation regime ($NO_x$-limited or VOC-limited). Here we leverage ground-based $O_3$ observations alongside space-based observations of $O_3$ precursors, namely $NO_2$ and formaldehyde (HCHO), to study the long-term

shifts in $O_3$ chemical regimes across global source regions. We first derive the regime threshold values for satellite-derived $HCHO/NO_2$ ratio by examining its relationship with the $O_3$ weekend effect. We find that a regime transition from VOC-limited to $NO_x$-limited occurs around 3.5 for $HCHO/NO_2$ with regional variations. By integrating data from four satellite instruments, including GOME, SCIAMACHY, OMI, and TROPOMI, we build a 27-year (1996 - 2022) satellite $HCHO/NO_2$ record, from which we assess the long-term trends in $O_3$ production regimes. A discernible global trend

towards $NO_x$-limited regimes is evident, particularly in developed regions such as North America, Europe, and Japan, with emerging trends in developing countries like China and India over the past two decades. This shift is supported by both increasing $HCHO/NO_2$ ratios and a diminishing $O_3$ weekend effect. Yet, urban areas still hover in the VOC-limited and transitional regime on the basis of annual averages. Our findings stress the importance of adaptive emission control strategies to mitigate $O_3$ pollution.

**1 Introduction**

Ozone ($O_3$) near the surface is an air pollutant with profound implications for human health and Earth's ecological system (Chiu et al., 2023; Nuvolone et al., 2018; Mills et al., 2016; Felzer et al., 2007). It is known to cause respiratory and cardiovascular diseases (Who, 2013). Chronic exposure to $O_3$ has been linked to an estimated 1.04 to 1.23 million premature mortalities globally in 2010, primarily due to respiratory ailments (Malley et al., 2017), and this issue of $O_3$-

related deaths has the potential to worsen despite the improvement of other air pollutants like fine particulate matter (Lu et al., 2018; Li et al., 2019; Wang et al., 2021). In addition to its harmful effects on human health, $O_3$ poses a threat to other species by inducing DNA damage in animals and affecting crop productivity and yield through disrupting the plant microflora (Manisalidis et al., 2020).

Ground-level $O_3$ is a secondary air pollutant, formed through photochemical reactions between oxides of nitrogen ($NO_x$)

and volatile organic compounds (VOCs). At ground level, $O_3$ formation is mostly in $NO_x$-limited regime globally, especially in rural or sparsely populated regions (Monks et al., 2015). In areas with high $NO_x$ emissions and relatively




low HCHO emissions, such as urban and metropolitan centers, $O_3$ formation can become $NO_x$-saturated, or in other words, VOC-limited. Freshly emitted NO, particularly from vehicular traffic, can locally deplete $O_3$ by reacting with it (Solberg et al., 2005), thereby curtailing $O_3$ accumulation in the immediate vicinity. Consequently, lower $O_3$ concentrations are

typically observed in urban areas (Paoletti et al., 2014; Simon et al., 2024). Over the past several decades, the evolution of global $O_3$ formation has been shaped by a complex interplay of socio-economic factors, including varying industrial activities and population movements, as well as environmental policies and changing climate (Pfister et al., 2014; Zhang et al., 2019). The combined effects of these factors are highly intricate. For instance, sustained declines in $NO_x$ and VOCs emissions have led to reductions in peak $O_3$ concentrations in many developed countries, but mitigating $O_3$ exposure at

the urban scale is still challenging owing to the nonlinearity of $O_3$-$NO_x$-VOC chemistry (Simon et al., 2016). Therefore, understanding the $O_3$ production regimes transition and its drivers are essential for devising effective mitigation strategies.

    $O_3$ sensitivity cannot be directly observed, which is often diagnosed through analyzing the relationship between observed $O_3$ and its precursors, or by using measurements of indicator species such as $NO_y$, formaldehyde (HCHO), reactive nitrogen ($NO_y$), hydrogen peroxide ($H_2O_2$), nitric acid ($HNO_3$) (Tonnesen and Dennis, 2000; Sillman, 1999, 1995).

However, ground-based measurements of these indicators are often limited, making satellite remote sensing a vital alternative for expanding the monitoring of these atmospheric species. Satellites provide retrievals of two key species: HCHO (Fu et al., 2007; Palmer et al., 2003), which is nearly proportional to the summed rate of VOC reactions with hydroxyl radicals (OH) and thus serves as an effective VOCs tracer (Sillman, 1995), and nitrogen dioxide ($NO_2$), which is prevalent in the boundary layer atmosphere and represent the majority of $NO_x$ (Duncan et al., 2010). The ratio of HCHO

to $NO_2$ (HCHO/$NO_2$) has been used to infer $O_3$-$NO_x$-VOC sensitivity (Choi et al., 2012; Jin and Holloway, 2015; Martin et al., 2004; Duncan et al., 2010; Jin et al., 2020; Jin et al., 2017). An important issue to use satellite HCHO/$NO_2$ is to determine the threshold values separating the $NO_x$-limited and VOC-limited regimes. Martin et al. (2004) and Duncan et al. (2010) use 1 and 2 regime threshold values, but follow-up studies show that the regime threshold values are uncertain (Jin et al., 2017; Souri et al., 2023; Wang et al., 2021; Schroder et al., 2017).

Over the past two decades, the global distributions of HCHO and $NO_2$ concentrations have been shaped by diverse emission reduction policies, resulting in distinct regional changes. In terms of $NO_2$, many anthropogenic regions have witnessed non-linear shifts or reversal years in $NO_2$ pollutant levels (Georgoulias et al., 2019). In developed regions such as the U.S. and European countries, substantial reductions in $NO_x$ emissions have been achieved, largely due to stringent national regulations (Russell et al., 2012; Krotkov et al., 2016; Toro et al., 2021; Gov.Uk, 2024) , whereas in developing

regions, $NO_x$ emission reductions have normally lagged behind. According to Zhao et al. (2013), there was a surge in $NO_x$ emissions in China until around 2010, after which a decline was observed. This decrease has been linked to technological advancements and the implementation of emission control measures in key industries (Sun et al., 2018). Given the diverse trends of $O_3$ precursor emissions, less is known how the $O_3$ production regime has changed over the past decades because of the emission changes. Here we aim to identify the long-term trends in satellite HCHO/$NO_2$ and

the reversal years in different region, which could signal a change in the direction of $O_3$ chemical regime changes.

    Another widely used method to characterize $O_3$ formation regimes is through comparing the weekend versus weekday



difference (WE-WD) in $O_3$ levels and its precursors. Under high $NO_x$, $O_3$ production rates paradoxically increase as $NO_x$ concentration fall; conversely, in scenarios with low $NO_x$, $O_3$ production rates decline. In most urban areas, characterized by high $NO_x$ levels, $O_3$ concentrations frequently display a significant rise on weekends relative to weekdays. Reasons for this "$O_3$ weekend effect" can be multifaceted and region-specific, involving reduced $NO_x$ concentrations altering VOC ratios, timing shifts in $NO_x$ emissions, increased VOCs and $NO_x$ emissions on weekend nights, and enhanced sunlight due to lower particulate matter emissions (Carb, 2003). This distinctive WE-WD $O_3$ pattern has been observed globally, first documented in New York City, U.S. (Cleveland et al., 1974), and subsequently reported in various regions including Europe (Sicard et al., 2020; Adame et al., 2014), East Asia: Tokyo, Japan (Yasuhiro et al.), the Pearl River Delta (Zou et al., 2019), the North China Plain (Wang et al., 2014), the Yangtze River Delta (Tang et al., 2008) and Taiwan (Tsai, 2005), North America: Mexico (Stephens et al., 2008) and whole U.S (Jaffe et al., 2022; Atkinson-Palombo et al., 2006), as well as major cities in Latin America: Santiago, Chile (Seguel et al., 2012) and Rio de Janeiro, Brazil (Martins et al., 2015). The varying $O_3$ weekend effect provides an opportunity to evaluate the chemical regimes of $O_3$ (Simon et al., 2024; Jin et al., 2020).

In this study, we aim to elucidate the long-term shifts in $O_3$ chemical regimes on a global scale using the two indicators: satellited derived $HCHO/NO_2$ ratios and ground-based observation of $O_3$ weekend effect. In Section 3.1, we examined the surface WE-WD $O_3$ concentration as a function of the tropospheric column $HCHO/NO_2$ ratio to identify the thresholds distinguishing different $O_3$ regimes. In Section 3.2, we analyzed the long-term trend of satellite-based $HCHO/NO_2$ and identified the trend reversals. These two steps set the stage for evaluating the long-term evolution of $O_3$ production regime. In Section 3.3, we synthesize above analyses with the objective of pinpointing the year when the $HCHO/NO_2$ ratio crossed the critical thresholds, indicating a shift from the VOC-limited to the $NO_x$-limited regime. The HCHO and $NO_2$ retrievals integrate 27-year (1996 - 2022) data from four satellite instruments: GOME/ERS-2, SCIAMACHY/ENVISAT, OMI/Aura and TROPOMI/Sentinel-5P. Finally, we assess whether the satellite-based trends in $O_3$ chemical regimes are consistent with long-term trends of the $O_3$ weekend effect in Section 3.4. Overall, our goal is to provide a view of $O_3$ regime changes across regions and decades, which could have implications for environmental policy and air quality management.

## 2 Data and Methods

### 2.1 Harmonized Satellite Retrievals of $O_3$ Precursors

We combine satellite retrievals of tropospheric $NO_2$ and HCHO vertical columns from four different satellite instruments, including: Global Ozone Monitoring Experiment (GOME), SCanning Imaging Absorption spectroMeter for Atmospheric CHartographY (SCIAMACHY) and Ozone Monitoring Instrument (OMI) and TROPOspheric Monitoring Instrument (TROPOMI). We use satellite-based products developed under the Quality Assurance for Essential Climate Variables (QA4ECV) project, which retrieves $NO_2$ and HCHO consistently using the same *a priori* profile obtained from TM5-MP (Boersma et al., 2018; Williams et al., 2017; De Smedt et al., 2017; Boersma et al., 2017b, a). The nadir resolution is $320 \times 40 \ km^2$ for GOME, $60 \times 30 \ km^2$ for SCIAMACHY, $24 \times 13 \ km^2$ for OMI and $5.5 \times 3.5 \ km^3$





for TROPOMI. The overpass time is around 10:00 AM local time for SCIAMACHY and GOME, ~ 1:30 PM for OMI and TROPOMI,

To build the relationship between observed $O_3$ weekend effect and satellite $HCHO/NO_2$, we mainly use OMI retrievals of HCHO and $NO_2$, as it provides the longest record with fine resolution suitable for studying the urban $O_3$ chemistry,

and the overpass time of OMI is well suited to detect the $O_3$ formation sensitivity during the afternoon as the $O_3$ photochemical production peaks and when the boundary layer is high and the solar zenith angle is small, maximizing the instrument sensitivity to HCHO and $NO_2$ in the lower troposphere (Jin and Holloway, 2015; Jin et al., 2017). To investigate the long-term changes in $HCHO/NO_2$, we construct annual average HCHO and $NO_2$ using tropospheric $NO_2$ and HCHO VCD data from the GOME (1996-2001), SCIAMACHY (2002-2003) and OMI (2004-2020) and TROPOMI

(2020 - 2022) datasets. GOME and SCIAMACHY and TROPOMI data are harmonized with reference to OMI data with a resolution of $0.25° × 0.25°$. The retrieval and harmonization scheme are described in Jin et al. (2020).

## 2.2 Ground-based $O_3$ observations

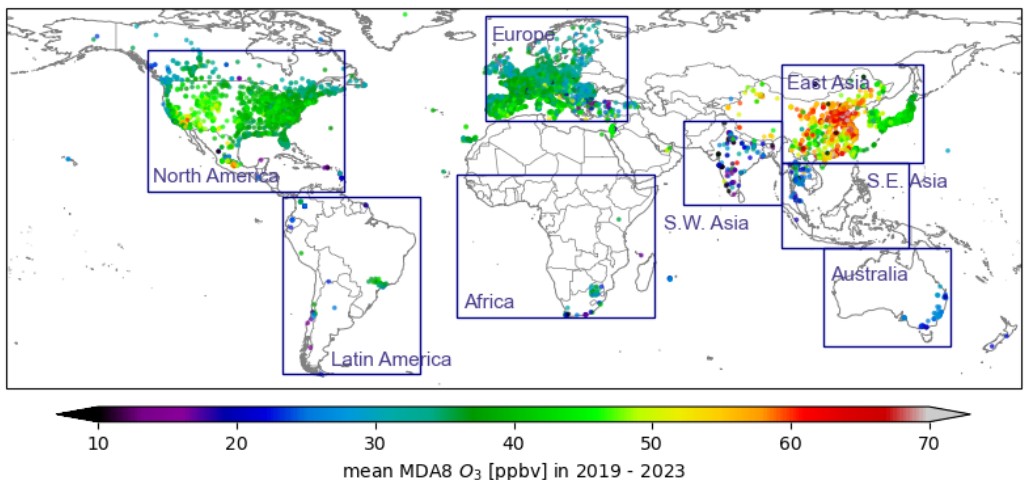

**Figure 1: Global distribution of $O_3$ (unit: ppbv) in the past 5 years (2019-2023). Data are sourced from the TOAR database.**

For $O_3$ data, we rely on TOAR-II database (https://toar-data.org/surface-data, last access: April 2024). Initiated by the Global Atmospheric Chemistry Project (GACP), TOAR has developed a cutting-edge database that provides hourly surface $O_3$ concentrations on a global scale since 1970 (Schultz et al., 2017), serving as an unparalleled resources for examining temporal trends in surface $O_3$ levels (Sicard et al., 2020). Notably, the observation records period various across monitoring stations, with earlier data in the U.S., Europe and Japan dating back to the 1970s-1980s, and later in

countries like the South Korea and Latin America, starting from 1995 to 2005. For China, South Africa, Southwest Asia, and densely populated Australian areas, records typically begin around 2015. To ensure rigorous study standards, we selected over 8700 stations with at least 5 consecutive years of data for our global analysis. Figure 1 illustrates the distribution of TOAR sites and main regions we focus on.



### 2.3 Building Connections Between Satellite HCHO/NO₂ and Ground-Based WE-WD O₃ Observation

We utilize monthly HCHO/NO₂ derived from the OMI to establish correlations with ground-based TOAR O₃ observation. We extract gridded daily OMI HCHO and NO₂ data (0.25˚ × 0.25˚) for days and grid cells with corresponding O₃ monitoring data, ensuring that both datasets are paired consistently in time and location. To quantify O₃ differences between weekends and weekdays, we designate Sunday to represent weekends and Tuesday to Thursday to represent weekdays, excluding other days to minimize carryover effects from typical workdays and rest-days. For each site and

weekly interval throughout the observation period, we calculated the mean differences in MDA8 O₃ concentrations (WE-WD O₃). Given the global scope of this analysis and the inherent complexity in defining distinct O₃ seasons across various regions, we utilize all-year data without seasonal selection. Using t-test at each site or grid to ascertain the statistical significance of WE-WD difference (p-value<0.05). For annual trends, we apply the non-parametric Mann-Kendall test (Kendall, 1975; Mann., 1945) coupled with Theil-Sen's slope estimator (Raj and Koerts, 1992; Sen, 1968). We examine

the annual trends in the WE-WD O₃ over 5-year rolling intervals to mitigate the effects of interannual meteorological variability (Pierce et al., 2010).

### 2.4 Detection of Long-term Trend Reversal in Annual HCHO/NO₂

As most regions show bi-directional trends of O₃ precursors, we hypothesize that a reversal of trend in HCHO/NO₂ can be found during our study period. To identify trend reversal years for the HCHO/NO₂ ratio at each grid point, we adopt

the method Georgoulias et al. (2019) used in the analysis of satellite-derived NO₂ trend reversals, originally adapted by Cermak et al. (2010) for studying solar radiation and global brightening trends. The approach is briefly described as follows:

Firstly, for each grid point and for each year *t*, a point score *S(t)* is calculated to quantify the potential for a trend reversal:

$$S(t) = \frac{\min(p(B_l),\ p(B_r))}{\text{abs}(B_l - B_r) \times \sigma_{B_{l+r}}} \tag{1}$$

Here, $B_l$, $B_r$ and $B_{l+r}$ represent the trends calculated over 5-year periods to the left [t - 4, t], right [t, t + 4], and spanning the year [t - 4, t +4], respectively. The 5-year interval is chosen to reduce the impact of interannual meteorology variability. $p(B_l)$ and $p(B_r)$ are the probabilities (p-value) of the trend B being statistically insignificant, while $\sigma_{B_{l+r}}$ signifies the error in trend fitting. The *p*-value of the hypothesis test, with the null hypothesis being a zero slope, using a Wald test with a *t*-distribution.

The time series data for each grid and period are fitted to a linear model:

$$Y_t = A + BX_t + N_t \tag{2}$$

where $Y_t$ is the annual mean value for year *t*, $X_t$ is time variable representing the year, *A* is the annual mean of the first year, *B* is the estimated slope of trend line, and $N_t$ represents the residual, or the discrepancy between the fitted and the





observed value.

A year is identified as a trend reversal year if it exhibits the lowest $S(t)$ value, an opposite sign between $B_1$ and $B_r$ ($B_1 \times B_r$ <0), and significant trend starts and ends (both $p(B_1)$ and $p(B_r)$ < 0.05). Selecting the year with the lowest $S(t)$ ensures a maximal difference in trends slope (max $|B_1 - B_r|$) on either side of the year, with the fitting error of the trend at this juncture, $\sigma_B$, being as pronounced as possible. This method is estimated to be capable of identifying reversal years with a very limited error of 0.5-1% and standard deviation between 2 and 5% (Cermak et al., 2010). The trend calculation, based

on data spanning 5 years before and after each year, helps to mitigate the impact of short-term extremes in pollutant concentrations, such as the dramatic decrease in emissions during the 2020 COVID-19 pandemic. This approach allows us to identify regions with long-term changes in trends.

## 3    Results and Discussions

### 3.1 Identification of regime thresholds for satellite HCHO/NO₂

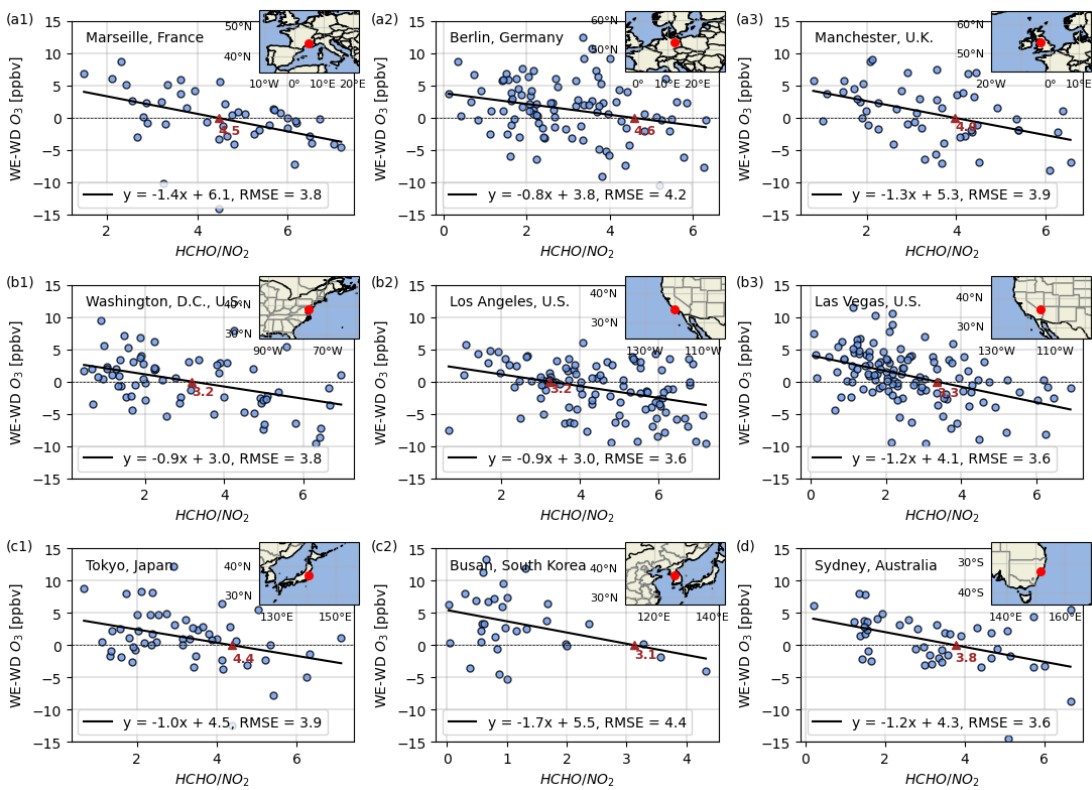


**Figure 2: Scatter plot of the monthly average satellite-derived HCHO/NO₂ ratio versus the WE-WD O₃ concentration in 9 representative urban sites from 2004 to 2022. The black line represents the fitted linear regression line. Red triangles denote the intersection points of the regression line with the WE-WD O₃ = 0 baseline.**

O₃ formation is a highly nonlinear process in relation to NO$_x$ and VOCs. When urban areas enter weekend, NO$_x$ emissions





typically decrease due to reduced commuting and industrial activities (Figure S1). In VOC-limited regime, $NO_x$ reduction leads to increased weekend $O_3$ levels (positive WE-WD $O_3$), whereas in $NO_x$-limited regime, it results in decreased weekend $O_3$ levels (negative WE-WD $O_3$). In theory, a transition threshold should exist between these two regimes. Figure 2 illustrates the correlation between monthly mean $HCHO/NO_2$ ratio from satellite data compared to the in-situ WE-WD $O_3$ in 9 representative metropolitan cities across 4 continents, showing a clear negative correlation and a transition from positive to negative WE-WD $O_3$ values at a specific $HCHO/NO_2$ ratio. Assuming that $O_3$ formation differences are

attributable to $NO_x$ changes only, $HCHO/NO_2$ value at the WE-WD $O_3 = 0$ crossing can be considered the threshold separating VOC-limited and $NO_x$-limited regimes. For example, in Washington, D.C., this key threshold is found to be 3.2 (Figure 2 b1).

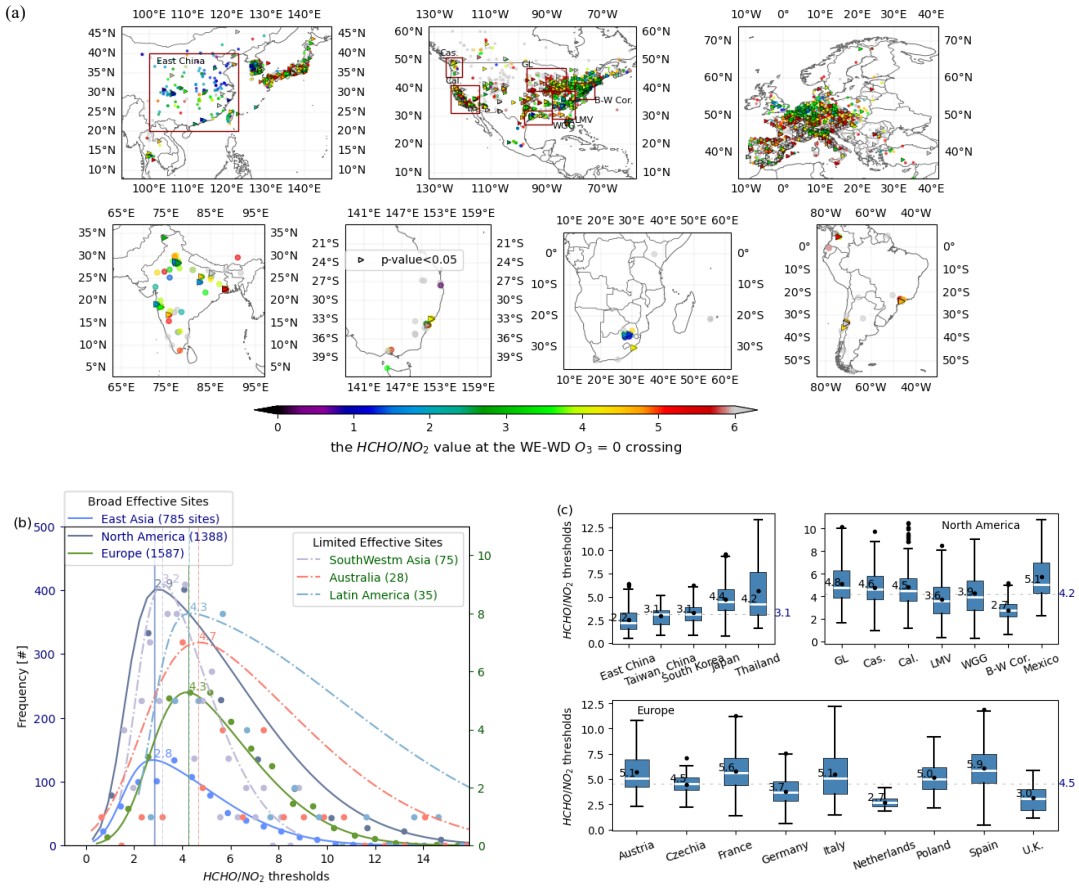

**Figure 3: (a) Global map of regime transitional threshold values for $HCHO/NO_2$, derived by assessing the correlation between**
**monthly $HCHO/NO_2$ and WE-WD $O_3$ difference. We restrict our analysis to ground sites with at least 5 years' observation. Triangles represent sites with linear regression lines fitting within the 95% confidence interval (Wald test for slope, p-value < 0.05). Specific economic regions are highlighted with red rectangles, including: Specific economic regions are highlighted with red rectangles, including: East China; Great Lakes region (GL), Cascadia region (Gas.), California (Cal.), Lower Mississippi Valley region (LMV), West Gulf Coast (WGG), and Boston-Washington Corridor (BW Cor.) in the U.S. (b) Distribution of site**
**numbers across varying threshold bins, with solid/dashed lines representing continents with >100 sites and 20-100 effective sites, respectively. (c) Box plot of thresholds in economically advanced regions of East Asia, North America, and Europe, as marked in (a). The dark-blue numbers on the right axis indicate the median threshold value for all regions presented.**





By aggregating all TOAR $O_3$ observations based on corresponding monthly OMI data, we evaluate the thresholds through linear regression between the monthly mean WE-WD $O_3$ and the $HCHO/NO_2$ ratio at the ground-based sites with at least

5 years' observations. The key thresholds indicating $O_3$ regime shifts are identified as the critical point where WE-WD $O_3$ changes sign. The spatial distribution and statistical results of the identified critical points are presented in Figure 3. Globally, robust linear relationships are observed, particularly pronounced in regions such as South Korea, Japan, the U.S., and Europe (Figure 3a), which also have the highest density of monitoring stations. We find a wide distribution of thresholds across different regions (Figure 3b), implying a large spatial variability in the threshold values. Among the

sites where the linear regression is statistically significant with $p$-value $< 0.05$, approximately 63% of the sites have threshold values between 2 to 5, with over 80% of these sites between 2.5 and 4.7, and the mean value around 3.5. East Asia has the lowest mean threshold value at 2.8 with the minimum over the East China (2.2) and maximum over Japan (4.4) (Figure 3c). In North America, the threshold value is around 2.9, with the eastern seaboard sites averaging 2.7±1 and the sites in western region, predominantly centered in California, slightly higher at 4.5±1, and the maximum value is

around 4.8, which is located in the southwest of the Great Lakes region. Europe, with the densest sites of robust linear relationship, has the second-highest critical threshold at 4.3, comparable to Latin America, and just below Australia's 4.7. In Europe, a lower threshold cluster from 2.5 to 3.5 is centered in western Germany, extending to Belgium, northeastern France, the Netherlands, and parts of eastern U.K., with similar low spots in northern Portugal, southern Spain, and northern Italy. Southwest Asia's values are centered around 3.2. Africa and Southeast Asia, with fewer than 20 effective

sites (over 5-year continuous observation), are excluded from the analyses due to limited representativeness.

It should be noted that these calculations do not account for the effects of short-term synoptic processes on temperature and the conditions affecting $O_3$ transport and diffusion. The regime thresholds have uncertainties, and previous studies typically assume a range for regime threshold values (Sillman, 1999; Jin et al., 2020; Jin et al., 2017). This implies that the critical threshold values identified in this study should not be considered definitive indicators that guarantee a regime

shift. Nonetheless, this method remains valuable for leveraging large-scale satellite data to track the global progression of $O_3$ regimes, especially in regions and periods where in-situ $O_3$ data are limited. We will further explore this using the statistically derived threshold in Section 3.3.

### 3.2 Long-term Trends and Trend Reversals in Satellite-based $HCHO/NO_2$

Using satellite-based $HCHO/NO_2$ as a determinant for identifying $O_3$ chemical regimes, we assess the spatial variations

and long-term evolution of $O_3$ chemical regime over global anthropogenic regions from 1996 to 2022. Figure 4 show the multi-year average $HCHO/NO_2$ ratio maps from 1996 to 2022, derived from the harmonized satellite dataset. Notable areas with extremely low $HCHO/NO_2$ ratios of below 1 include East China, Seoul-Suwon region in South Korea, the major urban areas of southern Honshu in Japan, and European regions centered around Belgium, the Netherlands, and eastern United Kingdom, as well as northern Italy. Local minima are found in metropolitan areas such as China's Pearl

River Delta, U.S. regions including Los Angeles and San Francisco, urban clusters along the East Coast, and South Africa's Johannesburg etc. These regions are likely under long-term VOC-limited regime. Regions with ratios below 2 include the extensive area of the eastern U.S., the Mumbai and Delhi-New Delhi corridor in India, and major Australian



cities like Canberra, Sydney, and Adelaide. The distribution of areas with low HCHO/NO₂ ratio close aligns with that of high NO₂ areas (Figure S2a). NOₓ emissions, primarily linked to population density and economic activities,

predominantly originate from high-temperature combustion processes involving nitrogen and oxygen, such as industrial emissions and vehicle exhaust (Liu et al., 2016). With its short atmospheric lifetime of a few hours to a day, the distribution of NO₂ reflects hotspots of power generation and fossil fuel consumption (Jamali et al., 2020). In contrast, HCHO, an intermediate in the degradation of various VOCs (De Smedt et al., 2015), exhibits a more uniform distribution due to the widespread biogenic sources of VOCs (Figure S2b).

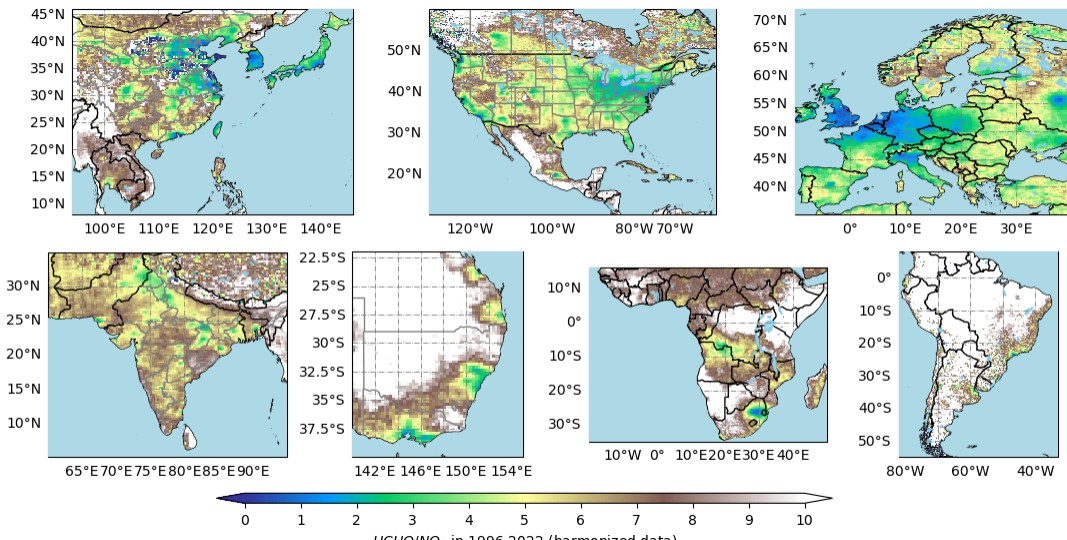


**Figure 4: Tropospheric HCHO/NO₂ ratio patterns using the self-consistent GOME, SCIAMACHY and OMI dataset for the combined period 1996 – 2022.**

Figure 5 shows the long-term linear trends of tropospheric HCHO/NO₂ ratios from 1996 to 2022. Here we focus on regions with long-term mean tropospheric NO₂ column greater than $1.5 \times 10^{15}$ molecules/cm² and statistically significant

trends with p-value < 0.05. Globally, satellite HCHO/NO₂ shows robust linear trends but with diverse directions. In densely populated area, we find positive trend in HCHO/NO₂ over developed nations and a downward trend in HCHO/NO₂ in developing regions. Significant positive trends, averaging $0.11 \pm 0.05$ yr⁻¹, are observed in Japan and extensive regions of the U.S., particularly along the eastern seaboard and in California. South Korea, the mid-south U.K., north France, Belgium, and part of the Netherlands, Germany, Spain and Italy exhibit a mild decline of $0.04 \pm 0.03$ yr⁻¹,

peaking at 0.12 yr⁻¹ in the urban cluster of northeast France. In contrast, central-eastern China, eastern India, show strong negative trends of $-0.1 \pm 0.05$ yr⁻¹, with the steepest declines seen in China's lower Yangtze River region (over -0.18 yr⁻¹) and Kolkata, India (-0.13 yr⁻¹). Weaker negative trends of less than -0.08 yr⁻¹ are also seen in eastern Europe near the Black Sea, Melbourne and Sydney, Australia and along northern Algeria's Mediterranean coast in North Africa.



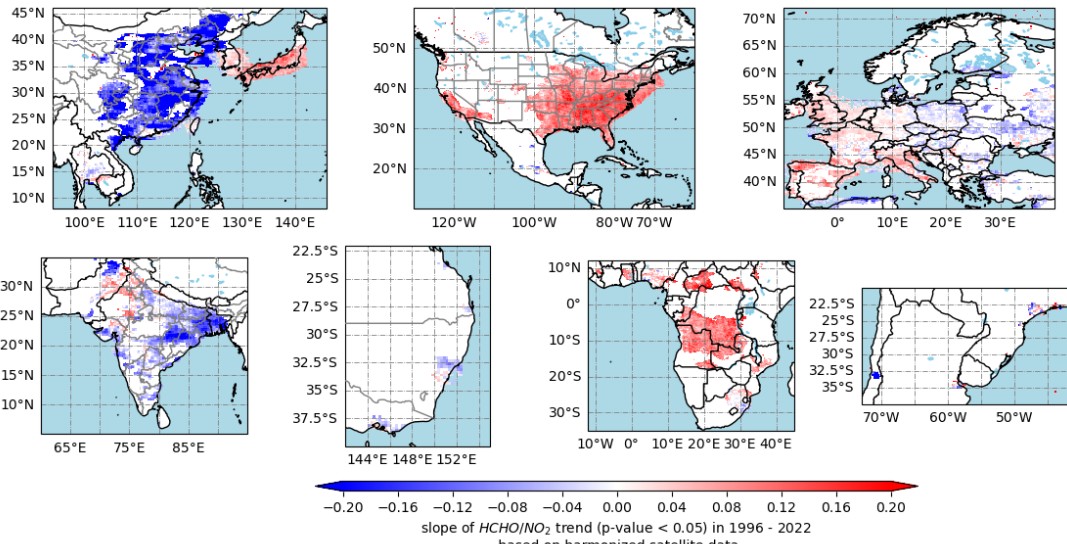

**Figure 5: Satellite-based trends of tropospheric HCHO/NO₂ ratios (1996–2022) for grids with a mean NO₂ VCD > $1.5 \times 10^{15}$ (molecules $\cdot$ cm$^{-2}$) and statistically significant trends at the 95 % confidence level.**

While we apply linear regression to identify the overall trends, trends in Figure 5 could shift due to factors like environmental policies and economic changes etc. These minor trend changes do not affect the linear fit's confidence. However, in certain areas, such as the North China Plain, significant turning points can render the linear fit insignificant. Here, rapid population growth and industrialization were followed by substantial policy-driven reductions in gaseous pollutant emissions, leading to trend reversals that the linear model fails to capture. This indicates that areas appearing blank in Figure 5 are not necessarily trendless but may have sharp turning points.

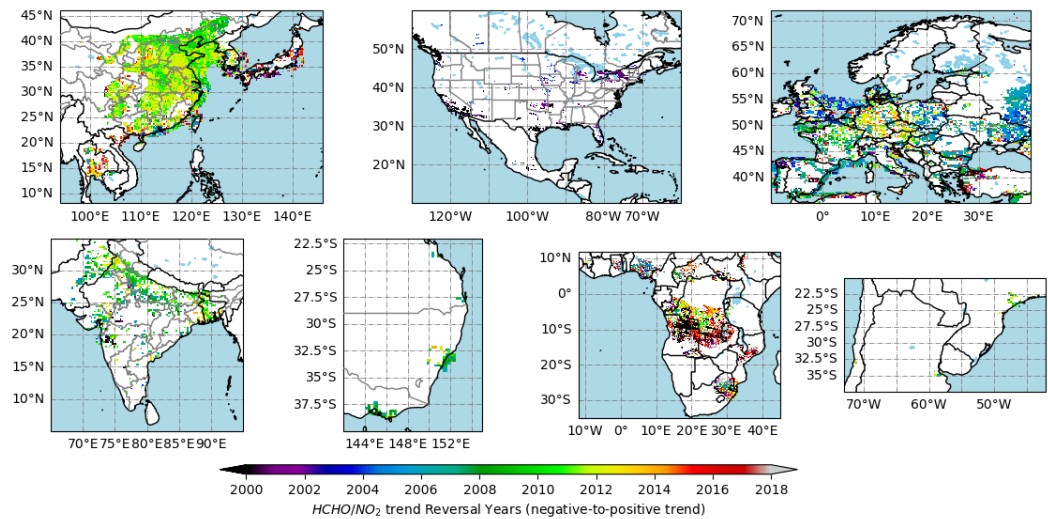

(a)





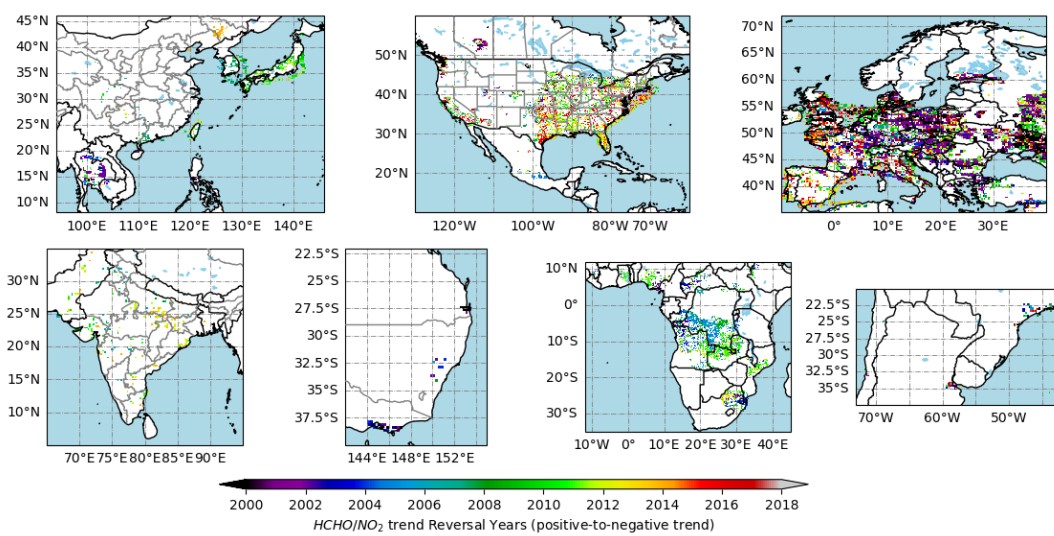

(b)

**Figure 6: The year of trend reversals of tropospheric HCHO/NO₂ ratio: (a) from negative to positive and (b) from positive to negative. Only grids with a with a long-term mean tropospheric NO₂ column greater than $1.5 \times 10^{15}$ molecules/cm² and statistically significant trends with p-value < 0.05 for the period before and after the year of reversals are shown.**

Next, we assess whether reversals of the trends exist in satellite HCHO/NO₂, using the methods described in Georgoulias et al. (2019). Figure 6 shows our estimated year when a persistent reversal of the trend occurs for HCHO/NO₂. We only include grid cells with statistically significant trends (*p*-value < 0.05) for both pre- and post-reversal periods. Figure 6a shows the regions where we find trend changes from negative to positive, and Figure 6b shows the regions where the trend transitioned from positive to negative. It should be noted that some regions may experience several reversals, here we only highlight the most significant trend reversals at each grid. The timing of reversals in the HCHO/NO₂ ratio is highly variable across different regions. In Asia, particularly in East China and North India, we observed a single reversal in the HCHO/NO₂ ratio from negative to positive trends around 2011. Conversely, major city clusters in North America exhibit a contrasting trend, with a significant shift from positive to negative trends occurring around 2012-2016. European countries generally display multiple distinct trend reversals. The first occurred in the early 2000s, transitioning from increase to decrease trends, followed by a shift back to positive trends around 2005-2012. Another notable shift to negative trends occurred in central and western European countries between 2015 and 2017. Major urban areas in southeast Australia saw a delayed single negative to positive reversal between 2006 and 2009. Overall, since the 21st century, there has been a notable upsurge in HCHO/NO₂ within the industrialized western world. Till 2015, the U.S. experienced a substantial growth rate of 52-124%, Europe by 12%-17%, and Japan by 77%. From 2015, a minor decline emerged in some U.S. regions, with California being one of the most pronounced, yet not widespread enough to form a significant regional pattern. This reversal correlates with the decline of HCHO and the leveling-off of NO₂ trends (Figure S3a). While meteorological factors can account for the 3-5 years cyclical fluctuations in HCHO/NO₂ ratios, the persistent changes in trends are primarily governed by variations in pollutant emission. In China, the post-2011 decline in HCHO/NO₂ ratio, aligned with the observed national NOₓ emission reductions, is largely attributed to NOₓ control measures in power plants



and industries and stricter vehicle emission standards (China IV) (Van Der A et al., 2017). Changes in VOCs emission
also contribute to this rapid increase; according to Liu et al. (2021), from 2011 to 2017, China's industrial VOCs emissions
increased from 11122.7 thousand tons/yr to 13397.9 thousand tons/yr. In the U.S., following the implementation of the
Clean Air Act Amendments in 1990, the EPA has placed a strong emphasis on the comprehensive assessment of pollutants,
moving from an focus on VOCs control to a more integrated strategy that addresses both VOCs and $NO_x$ in the formulation
of photochemical pollution control strategies (Amendments, 1990). This shift directly led to the significant reductions in
$NO_2$ levels across the U.S. since 2000, but the trends of HCHO are flat, largely due to the contributions from biogenic
VOCs. Between 2000 and 2015, $NO_2$ levels fell to ∼46% and HCHO levels to 92-98% of their initial values (Figure S3).
As a result, the long-term trend in HCHO/$NO_2$ over U.S. are dominated by the trends of $NO_2$.

Within the scope of our study period, $NO_2$ variations exceeded those of VOCs in the majority of the regions we interested.
For these regions, the HCHO/$NO_2$ trend is predominantly influenced by $NO_2$ trends, with VOC trends showing minor
impacts. A distinct negative inflection point observed in the $NO_2$ trend around 2011 in eastern China and in the early
2000s in the U.S. and Europe supports this (Figure S4). From 1996 to 2022, China's NCP region's $NO_2$ levels increased
fivefold, peaking at eightfold the initial value in 2011, while HCHO levels only rose to 1.5 times the initial value (Figure
S3). Europe also showed a more significant changes in $NO_2$ levels: Europe's $NO_2$ levels fell to 36-73% and HCHO to
120-133% of 1996 levels. For an in-depth understanding of the factors influencing HCHO/$NO_2$ trends, we refer readers
to key literatures on the long-term monitoring of HCHO and $NO_2$. Specifically, Georgoulias et al. (2019) provided a
satellite-based analysis of $NO_2$ trends from 1996 to 2017, which supports the long-term consistency of our observations,
albeit with minor discrepancies in the examined time periods. For a nuanced understanding of HCHO trends, both regional
studies (Fan et al., 2023; Kuttippurath et al., 2022) and global perspectives (De Smedt et al., 2008; De Smedt et al., 2015)
are instrumental.

Collectively, by combining linear fitting and reversals analysis, we discern a predominant positive global trend in the
HCHO/$NO_2$ ratio, or a shift from negative to positive in the past 27 years, with distinct regional variations. Developing
regions like East China and North India, which have relatively late economic development and pollution control measures,
typically show twice trend reversals. In contrast, developed regions such as North America and Europe, where
HCHO/$NO_2$ ratios began to increase significantly in the early 2000s exhibit a more consistent upward trend. Although
these developed areas have experienced minor downward reversals or stagnation in recent years due to the rate of $NO_x$
reduction slowing down.

### 3.3 Global Transitions in $O_3$ Chemical Regimes Delineated by Satellite-Derived HCHO/$NO_2$ Ratios

As elaborated in Section 3.1, the correlation between the HCHO/$NO_2$ ratio and the WE-WD $O_3$ response patterns show
similarities but also small variations across different regions. Here, for simplicity, we apply a uniform global threshold of
HCHO/$NO_2$=3.5 to explore potential years of $O_3$ regimes changes. To describe how the $O_3$ chemical regime has evolved,
we categorize into four main categories based on the long-term trends of HCHO/$NO_2$ (Table 1): (1) constant $O_3$ chemical
regimes: regions with a single VOC-limited or $NO_2$-limited regime status without regime transition during the study





period; (2) constant quasi regime: regions with single regime for most of the time but has potential to exceed the threshold value of 3.5; (3) single shift of the regime: regions with single shift, either from VOC-limited to $NO_x$-limited, or vice
versa; (4) multiple shifts of the regime: regions with nonlinear trends in $HCHO/NO_2$, in which a reversal of the trends in $HCHO/NO_2$ is identified, and $HCHO/NO_2$ crosses over the threshold values at least once. This classification takes into account the initial conditions of $O_3$ regimes and their transitional characters based on the observed $HCHO/NO_2$ trend. Here, we examine the $O_3$ regime changes based on annual average $HCHO/NO_2$, but the $O_3$ chemical regime should vary seasonally (Jin et al., 2017; Jacob et al., 1995), typically becoming more $NO_x$-saturated in wintertime and more $NO_x$-
limited in summertime. We do not account for the seasonality of $O_3$ production regimes because the definitions of seasons vary by climate regions would complicate the comparisons across different regions.

**Table 1: Criteria for distinguishing different types of O₃ regime transitions.**

| Category | Subcategory | Description of HCHO/NO₂ Trend Pattern | Transition Year |
|---|---|---|---|
| Constant regimes | 1.1 fixed NOₓ-limited | Typical NOₓ-limited condition (HCHO/NO₂≥ 4.5) for 80% of the period. | |
| | 1.2 fixed VOC-limited | Typical NOₓ-saturated condition (HCHO/NO₂≤ 2.5) for 80% of the period. | |
| Constant quasi regimes | 2.1 fixed quasi-NOₓ-limited | HCHO/NO₂ ratio within 3.5 - 4.5 for 80% of the period, with the possibility of exceeding 4.5 but never falling below 3.5. | No regime transition |
| | 2.2 fixed quasi-VOC-limited | HCHO/NO₂ ratio within 2.5 - 3.5 for 80% of the period, with the possibility of falling below 2.5 but never exceeding 3.5. | |
| Linear trend with regime transition | 3.1 shift from VOC-limited to NOₓ-limited | positive HCHO/NO₂ trend (↗) and crosses the HCHO/NO₂ = 3.5. | |
| | 3.2 shift from NOₓ-limited to VOC-limited | negative HCHO/NO₂ trend (↘) and crosses the HCHO/NO₂ = 3.5. | |
| Non-linear trend with regime transition | 3.3 shift from VOC-limited to NOₓ-limited | For negative to positive trend (↘↗): HCHO/NO₂ crosses 3.5 after the reversal year<br>For positive to negative trend (↗↘): HCHO/NO₂ crosses 3.5 before the reversal year | Intersection with HCHO/NO₂ = 3.5 |
| | 3.4 shift from NOₓ-limited to VOC-limited | For negative to positive trend (↘↗): HCHO/NO₂ crosses 3.5 before the reversal year | |



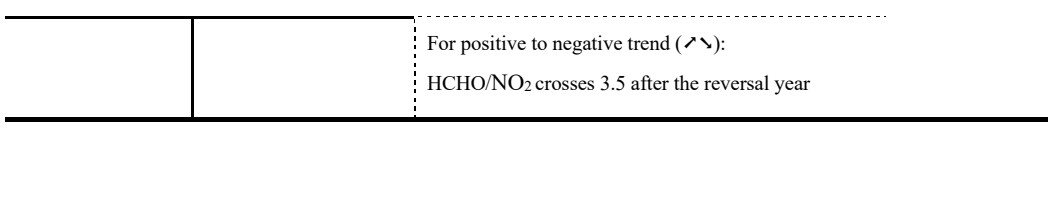

For positive to negative trend (↗↘):

HCHO/NO₂ crosses 3.5 after the reversal year

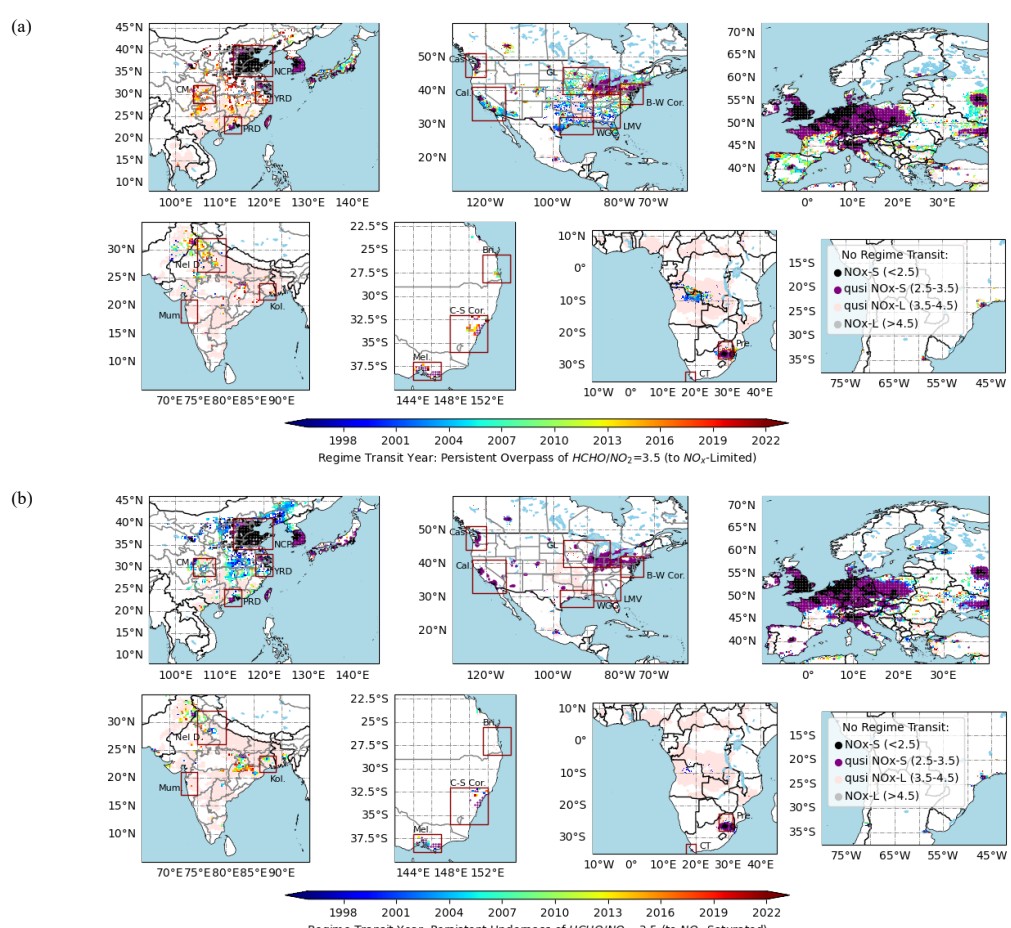

**Figure 7: Estimated transition year of O₃ regimes: grids where the annual HCHO/NO₂ ratio exceeds 3.5 (a) with a positive annual trend, and (b) with a negative annual trend. Only grids with a with a long-term mean NO₂ VCD > $1.5 \times 10^{15}$ (molecules · cm⁻²) are shown. Specific economic regions are highlighted with red rectangles, including: Specific economic regions are highlighted with red rectangles same as Figure 3a, adding: North China Plain (NCP), Yangtze River Delta (YRD), Pearl River Delta (PRD), and Chongqing Municipality (CM) in China; New Delhi (Nel D.), Mumbai (Mum.), and Kolkata (Kol.) in India; Melbourne (Mel.), Capital-Sydney Corridor (C-S Cor.), and Brisbane (Bri.) in Australia; Pretoria (Pre.) and Cape Town (CT) in South Africa.**

Figure 7 shows the classification of the regime changes. Globally, HCHO/NO₂ has shown a general upward trend post-2011, yet many central economic zones remain VOC-limited or quasi-VOC-limited, with the HCHO/NO₂ ratio not exceeding the threshold of 3.5 (Figures 7 and 8). Key regions, including China's NCP, YRD and PRD, Seoul in South Korea, Japan's metropolitan areas, the U.S. East Coast and California's Los Angeles and San Francisco areas, parts of





center-western Europe, India's New Delhi, South Africa's Pretoria and Cape Town region have been consistently VOC-limited for the majority of period (over 80%) since 1996. India, with its unique climate and emissions profile, exhibits a large area in the quasi-$NO_x$-limited range (3.5-4.5). India's hot and humid climate contributes to elevated BVOC and HCHO emissions (Kuttippurath et al., 2022), reducing $NO_x$ lifetime and accumulation, which tends to be more $NO_x$-limited. We observe an urban-rural transition from VOC-limited to $NO_x$-limited in megacity clusters, with the periphery

zones showing the initial shift toward $NO_x$-limited, progressing inward. This pattern is particularly evident in megacities such as Tokyo, San Francisco and Los Angeles. Regions have extensively transitioned to the typical $NO_x$-limited stage include the middle parts of Honshu in Japan, south California, parts of eastern U.S., and various regions in Europe.

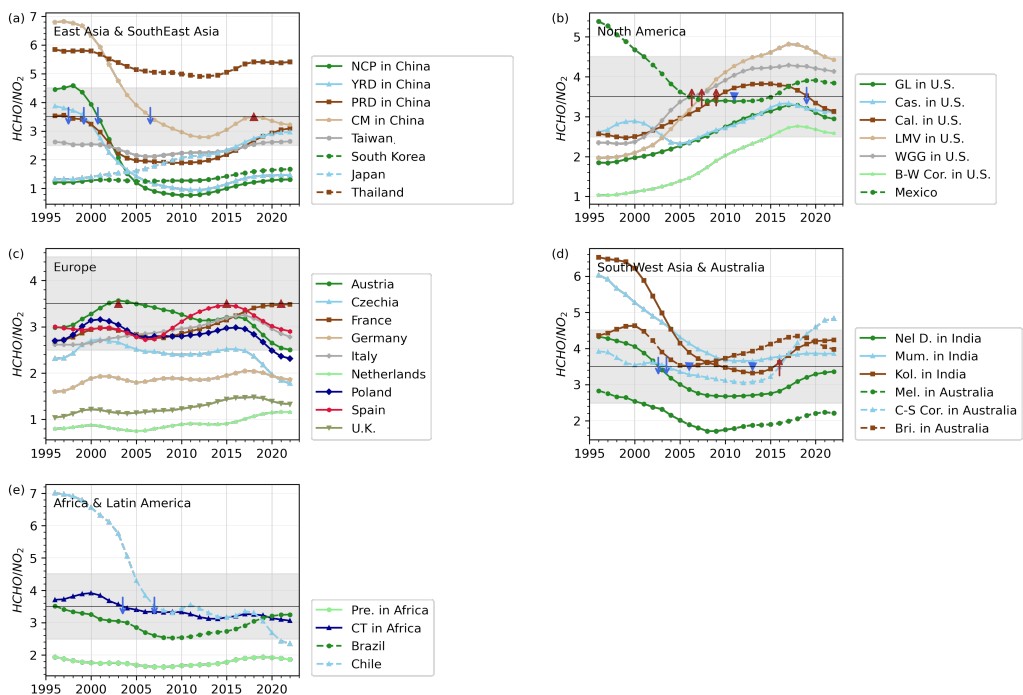

**Figure 8: Long-term trends of tropospheric HCHO/$NO_2$ ratio across major economic regions marked in Figure 7 based on the**
**harmonized satellite data. Here we only focus on grid cells with long-term mean tropospheric $NO_2$ column greater than 1.5 ×**
**$10^{15}$ molecules/cm$^2$. The arrow lines depict the intersection points with HCHO/$NO_2$=3.5.**

Figure 8 shows the long-term evolution of the satellite-based HCHO/$NO_2$ in different sub-regions of the world. The temporal evolution of regional $O_3$ regimes is intricately tied to a variety of emission policies, economic development, and geographical characteristics, such as surface coverage and regional meteorological conditions. Nations that embarked on
industrialization and VOCs reduction policies early on, like the U.S., Japan, and European countries, were predominantly in VOC-limited regime (HCHO/$NO_2$<2.5) by 1996 and have since shown a significant increasing trend in the HCHO/$NO_2$ ratio, particularly post-2000. Although most regions in the U.S. and some European countries (e.g., France, Germany, and Italy etc.) have reached peak levels around 2015-2017 and are currently experiencing a plateau or modest decline in the HCHO/$NO_2$ ratio, this has not significantly altered the overall upward trajectory of the ratio over the past 27 years. A





few regions have surpassed critical thresholds to $NO_x$-limited conditions—most notably observed in the southeast U.S. between 2006 and 2010—while most have yet to reach the 3.5 thresholds, either remaining in the typical VOC-limited (e.g., South Korea, China's NCP, YRD, etc.) or $O_3$ transitional regime (e.g., most regions in the U.S.). In regions that have attained $HCHO/NO_2 > 3.5$, such as the LMV, WGG, and California in the U.S., the post-peak decline did not cause a secondary transition point causing an $O_3$ regime shift. California, for example, surpassed the key threshold around 2009

and reached a maximum value within the quasi-$NO_x$-limited range by 2014, showing a re-crossing of the threshold around 2018, but still within the transitional regime.

Economies with relatively late industrial takeoffs, such as those in East China and India, initially had a high $HCHO/NO_2$ ratio around 1996, indicating $NO_x$-limited condition. As these economies bloomed in the late 1990s, $NO_x$ emissions from fuel combustion caused a sharp decline in the $HCHO/NO_2$ ratio, particularly in 2000-2005 period. In eastern China, the

NCP and YRD regions saw a rapid decrease in the $HCHO/NO_2$ ratio (-0.56/year), dropping from ~4 in 2000 to ~1.2 in 2005 (Figure 8a). Post-2005, the decline rate slowed, and by 2011, a slow reversal (<0.1/year) began, but it still remains in a VOC-limited regime to date. In contrast, Indian cities like Mumbai and Kolkata experienced a rapid decline in the $HCHO/NO_2$ ratio since 2000, but have not significantly crossed the key threshold (Figure 8d). In the Southern Hemisphere, regions like Australia, Africa and Latin America, typically exhibit low $NO_x$ emissions ($< 1.5 \times 10^{15}$ molecules/cm$^2$) across

vast areas due to the highly uneven population distribution. Melbourne's $HCHO/NO_2$ trend reversed around 2008, varying within a VOC-limited range of 1.8 to 2.8. The Canberra-Sydney corridor and Brisbane area show higher ratios (3.1 to 4.8) with no regime change throughout the observation period (Figure 8d). In Africa, the Pretoria region and Cape Town in South Africa exhibit a slow annual decrease of <-0.02/year (Figure 8e), averaging around 1.8 and 2.8. Latin American megacities have also remained in a transitional regime over the past two decades. These observations highlight the

complexity of $O_3$ regime patterns and the significant influence of region-specific factors.

Note that the regime threshold values are subject to large uncertainties due to factors such as meteorological conditions and satellite detection noise (Jin et al., 2017; Souri et al., 2017; Souri et al., 2020; Schroeder et al., 2017). The key $HCHO/NO_2$ threshold of 3.5 used here is derived from observation sites from TOAR network, which tend to be in urban or accessible areas, which is more reflective of regions with significant human impact rather than pristine natural

environments. These factors have the potential to bias the estimated transition year. However, such bias is not significant. It is estimated that a ~10% variation in the threshold (e.g., from 3.5 to 3.9) would shift the estimated transition years by only about 1-2 years for the major global regions. The uncertainty introduced by this simplified approach is considered acceptable and does not compromise the analysis's effectiveness in pinpointing critical transition years.

**3.4 $O_3$ Regime Transition Consistent with Diminishing $O_3$ Weekend Effect**

Over the past 2 decades, a significant reduction in $NO_2$ concentrations is found (Figure S3). Despite this trend, weekend $NO_2$ levels remain lower than those on weekdays (WE-WD $NO_2 < 0$) in most areas, although this $NO_2$-weekend-low phenomenon is weakening (Figure S1). Differential $NO_x$ saturation levels influence $O_3$ sensitivity to $NO_x$, resulting in varying WE-WD $O_3$ patterns. The interannual evolution in these patterns, in terms of sign and magnitude, provides another



piece of evidence of O₃ regimes changes. Figure 9 shows a spatial overview of the WE-WD O₃ differences across three
distinct periods (2004 – 2013, 2014 – 2018, 2019 - 2023) over the past two decades. Figure 10 shows the long-term O₃
weekend effect averaged across the sites. For robust statistical analysis, continental regional-averaged trends are limited
to sites with at least 10 years of data, focusing on East Asia, North America, Europe, and Latin America.

**Figure 9: Two-decade evolution of WE-WD O₃ differences across three distinct period: (a) 2004-2013, (b) 2014-2018, (c) 2019-2023. Significant (p-value of t-test < 0.05) WE-WD O₃ difference and WE-WD NO₂ differences are denoted by triangles and**





**black-edged symbols respectively.**

**Figure 10: Long-term evolution WE-WD O₃ difference derived from TOAR ground-based O₃ observation. Only sites with >=10 years of data are included in the continental region-averaged statistics (black line), and sites with >=3 years of data are included in the major economic regions and selected countries statistics.**

The WE-WD O$_3$ trend is inversely correlated with the HCHO/NO$_2$ ratio, showing an initial rise followed by a decline

        since 1980, with meteorological-driven interannual fluctuations observed throughout the period. The inflection points

        vary across different regions, generally occurring within the decade between 1995 and 2005 (Figure 10). The Theil-Sen

        slope, calculated from the most recent two decades of data, reveals a significant and widespread negative trend in WE-



WD $O_3$ concentrations at the majority of global monitoring sites, with a robust trend significance (p-value <0.05)
confirmed by the Mann-Kendall test, especially in North America, Europe, Japan, South Korea, and developed regions
of Australia and Latin America (Figure S4). In Europe, before 2013, over 70% of TOAR sites demonstrated significant
weekend $O_3$ increases (Figure 9a); by 2019-2023, only 2% of sites continued to show significant weekend elevations
(Figure 9c). The ensembled time series of Europe sites indicates that the peak WE-WD $O_3$ occurred around 2005, at
approximately $2.5 \pm 1.1$ ppbv, whereas by 2023, it had significantly decreased to near zero (Figure 10c). Similar reductions
have been observed in the U.S., with the largest WE-WD $O_3$ occurring between 1995 and 2000, and the most substantial
differences noted in California around 1998, peaking at approximately 4 ppbv (Figure 10b). In East Asia, where long-
term $O_3$ observations spanning over a decade are confined to Japan and South Korea, the aggregated data from monitoring
sites indicated a notable annual maximum around 1999, registering $3.3 \pm 0.5$ ppbv (Figure 10a1). This value is more
reflective of Japan's data, given that South Korea's monitoring began in 2000, with its own maximum occurring around
415   2012.

In Latin America, developed countries such as Brazil and Chile observed their peaks around 2005, with values around 4
ppbv (Figure 10d). In China and India, extensive ground-based $O_3$ monitoring networks were established within the last
decade. Eastern China, encompassing the NCP and YRD super urban agglomerations, aligns with the global trend of
reduced weekend $O_3$ (Figure 10a2). In India, however, rapidly developing areas such as New Delhi, Mumbai, and Kolkata
are experiencing a rise in WE-WD $O_3$, indicating a complex interaction between emissions and atmospheric dynamics
(Figure 10e). Note that due to short observation periods and moderate statistical significance (p-values between 0.05 and
0.1 from the Mann-Kendall test) (Figure S5), results from China and India are best used for qualitative analysis only.

By 2023, a few regions show a reversal of $O_3$ weekend effect on annual basis. The U.S. LMV area, which first showed a
WE-WD $O_3$ transition to negative values, has kept WE-WD $O_3$ levels around $-1 \pm 0.5$ ppbv since 2010 (Figure 10b).
France and Spain have also shown slightly higher weekend $O_3$ levels from 2020, but not significant (Figure 10c). Sporadic
sub-zero WE-WD $O_3$ in areas such as parts of California, the BW and WGG regions in the U.S. indicate a potential move
towards $NO_x$-limited conditions, though inconsistent.

Overall, the $O_3$ weekend effect's long-term changes are consistent with the weakening VOC-limited conditions outlined
in Section 3.3, with most regions classified as VOC-limited or in transition. This is evident in the fact that, in most regions,
weekend $O_3$ levels are slightly higher than or equal to those on weekdays. These results are consistent with the satellite-
derived $HCHO/NO_2$ observations, indicating a still ongoing transition in the $O_3$ chemical regime.

**4 Conclusion**

In this study, satellite-derived $HCHO/NO_2$ ratios and ground-based $O_3$ observations were directly connected to capture
the nonlinearity of global shifts in $O_3$ chemical regimes. Key findings are as follows:

The evolution of $O_3$ regimes is discernible through the analysis of $HCHO/NO_2$ ratio and WE-WD $O_3$ trends. We have



pinpointed distinct regional thresholds—2.8 for East Asia, 3.2 for Southwest Asia, 3.0 for North America, 4.3 for Europe and Latin America, and 4.7 for Australia—that signify the pivotal transition points in $O_3$ regimes. These thresholds are shaped by variations in regional energy profiles, meteorological patterns etc.

Amidst the ongoing changes in the ratios of $O_3$ precursors, a global trend towards $NO_x$-limited $O_3$ regimes have emerged over the past two decades. This is evidenced by both the rising $HCHO/NO_2$ ratios and the diminishing $O_3$ weekend effect, particularly in densely populated regions. Applying linear fitting and reversals analysis, we've observed a predominant positive or negative-to-positive shift global trend in the $HCHO/NO_2$ ratio over the past 27 years. Later-industrializing regions like East China and India initially saw a decline before rebounding around 2011; industrialized nations like the U.S., Europe, and Japan experienced significant increases in the $HCHO/NO_2$ ratio from the early 2000s due to substantial

$NO_x$ emission reductions. However, by 2023, most urban regions' annual-mean $HCHO/NO_2$ ratios have not significantly surpassed the 3.5 threshold, indicating they remain within VOC-limited or transitional regimes. However, $O_3$ chemical regime varies seasonally, and we expect the regime transition has occurred during the warm season when $O_3$ pollution is highest. Regarding WE-WD $O_3$, while some regions like France and northern Spain show lower weekend levels, the majority still report slightly higher weekend $O_3$ on annual basis, but not statistically significant anymore. A few areas,

such as the southeastern U.S., heavily influenced by BVOCs, have clearly entered an $NO_x$-limited regime on the annual basis. These results align with the general trend of weakening VOC-limited conditions。

The transitional zone for $O_3$ regimes should be a range rather than a fixed $HCHO/NO_2$ ratio. Our study simplifies this by using the central value of this range, acknowledging a limitation. Despite this, our findings provide valuable insights, highlighting the need for adaptable emission controls in response to atmospheric changes. Early industrialized nations

could benefit from policies addressing both $NO_x$ and VOCs to further curb $O_3$, while later industrializers should prioritize $NO_x$ controls to prevent excessive $O_3$ formation.

**Acknowledgement**

Support for this project was provided by NASA Aura Science Team and Atmospheric Composition Modeling and Analysis Program (grant number: 80NSSC23K1004). This research used the computational cluster resource provided by

the Office of Advanced Research Computing (OARC) at Rutgers, The State University of New Jersey. We are grateful to the many scientists who contributed to the GOME, GOME-2, SCIMACHY, OMI and TROPOMI instruments and products.

**Data Availability**

Multi-satellite products (GOME, SCIMACHY, OMI) of tropospheric $NO_2$ and HCHO vertical columns are developed

under the EU FP7-project Quality Assurance for Essential Climate Variables (QA4ECV) are publicly available at https://knmi.sitearchief.nl/?subsite=qa4ecv#archive. TROPOMI $NO_2$ data (https://doi.org/10.5270/S5P- s4ljg54) and TROPOMI HCHO (https://doi.org/10.5270/S5P-vg1i7t0) are available from NASA Goddard Earth Sciences (GES) Data



and Information Services Center (DISC, https://disc.gsfc.nasa.gov/datasets/ ). Ground-based $O_3$ observations are available from TOAR-II database (https://toar-data.org/surface-data). The harmonized annual satellite-based HCHO and
$NO_2$ products will be made publicly available at the publication stage.

**Author contribution**

**Y.T.:** Methodology, Formal Analysis, Investigation, Visualization, Writing - Original Draft. **S.W:** Data Collection of Satellite $NO_2$ products. **X.J.:** Conceptualization, Supervision, Methodology, Data Curation, Funding Acquisition, Writing - Review & Editing. All authors have given approval to the final version of the manuscript.

**Competing interests**

The authors decline that they have no conflict of interest.

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
