# Peer review of "Global Patterns and Trends in Ground-Level Ozone Chemical Formation Regimes from 1996 to 2022"

_EGUsphere, 2025_

## Author Comment (AC1)

**Reply to Reviewer 1**

*This manuscript focuses on identifying global surface ozone formation regimes and their long-term trends. The authors use satellite observations of HCHO/NO$_2$ ratio, together with ground-based ozone observations, to determine whether a region is NOx-limited, VOC-limited, or in a transitional regime. The main findings are (1) The HCHO/NO$_2$ ratio can be linked to the ozone weekend-weekday effect (WE-WD). (2) Satellite and surface observations indicate similar long-term changes in regimes across different continents. (3) There has been a global transition from a VOC-limited regime toward a NO$_x$-limited regime, and most urban areas remain VOC-limited.*

*This study is valuable in demonstrating the effectiveness of this satellite indicator by linking it to the surface WE-WD ozone effect and identifying long-term trends in global ozone chemical regimes. However, I find the paper not well organized, with some sections being repetitive and the analysis lacking depth. Therefore, this manuscript requires thorough editing before being considered for acceptance in ACP.*

Reply: We would like to thank the reviewers for their time to review this paper. We have revised the manuscript following the reviewers' suggestions. Below are our point-by-point responses to the comments, along with the corresponding revisions.

***Major issues:***

1. *In Section 3.1, the authors define HCHO/NO2 thresholds at each site and find large spatial variability across regions. However, in their later analysis, they use a uniform threshold everywhere, which could lead to misidentification of regimes. An example of potential misidentification: in the last section (Line 450), where they conclude that, unlike other places, the southeastern US has entered an NO$_x$-limited regime. However, this could simply be because this region requires a higher threshold than 3.5 (as shown in Figure 3a). I would recommend using at least region-specific thresholds, as derived in Figure 3c.*

Reply: That's a good point. In the revised manuscript, we have applied region-specific thresholds to classify O$_3$ production regimes (Figures 9 and 10). We have accordingly updated all relevant discussions in Sections 3.3 and 3.4 of the revised manuscript. This modification particularly enhances the interpretation of regional variations. Please see the relevant revisions in Section 3.3 and Section 3.4 of the revised manuscript.

[Figure]

Figure 9: Temporal evolution of tropospheric HCHO/NO₂ ratios by region, with symbols indicating annual O₃ regimes: VOC-limited (solid circle), transitional range (open circle), and NOₓ-limited (open triangles). Regional peak threshold values ($\tilde{x}$, marker on y-axis) correspond to **Error! Reference source not found.**c definitions.

[Figure]

Figure 10: Spatial distribution of estimated transition years for O₃ regime shifts, based on annual HCHO/NO₂ ratios crossing region-specific thresholds (identified in **Error! Reference source not found.**b).

2. *The analysis should be more in depth. The authors provide little interpretation of (1) the large regional variability in the HCHO/NO₂ threshold (e.g., what are the key driving factors, and why is the threshold in East China much lower than in other countries), and (2) the drivers of changes in HCHO, NO₂, and the HCHO/NO₂ ratio outside of East Asia and the US, including Europe, India, Australia, Africa, and South America (e.g., why is the HCHO/NO₂ trend in Africa significantly positive). This lack of discussion makes it seem as though the authors are only familiar with the background and policies of East Asia and the US, despite the study's intended focus on global changes.*

Reply: We acknowledge the importance of understanding these underlying causes and would like to follow the reviewer's suggestion to deepen analysis of regional threshold variability. We have added discussions on the potential causes of the spatial variability of the regime thresholds in the revised manuscript:

The variations of the regime threshold values of HCHO/NO$_2$ are likely caused by several factors. First, here we use tropospheric column HCHO/NO$_2$ to represent the near-surface O$_3$ chemistry, which is affected by the relationships between column and surface HCHO and NO$_2$ (Jin et al., 2017). The column-to-surface relationship is determined by the boundary layer height and the vertical profiles of HCHO and NO$_2$, which should vary spatially (Adams et al., 2023; Zhang et al., 2016b). Second, HCHO is used as an indicator of VOCs, but the yield of HCHO from oxidation of VOCs varies with different species (Shen et al., 2019; Chan Miller et al., 2016; Zhu et al., 2014). Regions dominated by biogenic VOC emissions like southeast U.S., tropical regions generally have larger HCHO yield (Wells et al., 2020; Palmer et al., 2007; Palmer et al., 2006). Third, the local chemical environmental may also differ spatially. For example, the lower thresholds in China are consistent with elevated regional NO$_x$ levels (Jamali et al., 2020) and enhanced secondary aerosol formation in this region, which may promote radical loss (Li et al., 2019; Liu et al., 2012). Here we use statistical methods to derive the regime thresholds. Further attribution of the spatial variations is beyond the scope of this study, which warrant further investigation.

We also have also expanded discussions of the drivers of the changes in HCHO/NO$_2$ in global regions in the revised manuscript:

**3.2.1 Spatial Patterns and Linear Trends**

……

The observed trends mirror the global redistribution of O$_3$ precursors since 1980, where developed nations achieved emission reductions while developing Asia - particularly Southeast, East and South Asia - experienced dramatic increases (Zhang et al., 2016a). Specifically, Europe and North America achieved >60% NO$_x$ reductions between 1990-2022 through stringent air quality policies, while Asia experienced an 86% increase during the same period, with India's emissions nearly tripling to 9.4 million metric tons by 2022 (https://www.statista.com/, note as Statista Data from here). Although VOC reductions in developed regions (e.g., -46% in the U.S. since 1990 to 2023, Statista Data) would theoretically drive HCHO/NO$_2$ ratios downward, the observed trends are primarily governed by NO$_x$ dynamics, as evidenced by the stronger correlation between NO$_2$ and HCHO/NO$_2$ trends compared to HCHO alone (Figure S3). Local-scale variations in either VOCs or NO$_x$ may account for regional deviations from these general patterns. Japan represents a unique case in East Asia, showing robust linear growth in HCHO/NO$_2$ ratios, resulting from its early and rigorous regulatory framework targeting both mobile and stationary sources. The country's vehicle emission controls, initiated in 1966 (initially CO-focused), evolved through progressive NO$_x$ standards for light-duty vehicles (1973), stricter gasoline/diesel limits (1989), and world-leading regulations by 2003 (NO$_x$ regulations surpassing contemporaneous U.S. and European standards) (https://www.env.go.jp/air/). Parallel industrial policies revised stationary source NO$_x$ limits four times since 1973. These measures drove a 33.7% reduction in national NO$_x$ emissions from 2005 to 2014 (1.93 to 1.28 million metric tons; Statista Data), positioning Japan's emissions at merely 5.8% of China's regional total (0.68 vs. 11.76 Tg N/yr; Han et al., 2020). Satellite data corroborate a 27-year decline in tropospheric NO$_2$

columns.

Over the African continent, almost no change is detected, with one notable exception - the Congo Basin. This finding aligns with previous satellite observations showing no significant trends in either $NO_2$ (Hilboll et al., 2013) or HCHO (De Smedt et al., 2015) over most of Africa. However, existing studies have not sufficiently explained the continental disparity between Africa's overall trend stability and the Congo Basin's unique behavior. The Congo Basin presents a particularly intriguing case, exhibiting anomalously strong negative HCHO/$NO_2$ trend (Figure 5) and complex identified reversals (**Error! Reference source not found.**). This region's distinct atmospheric chemistry likely stems from its status as one of the world's most active biomass burning hotspots, where competing environmental factors may drive the observed anomalies: a global reduction in burned area including Africa (1998-2015; Andela et al. (2017)) versus persistent localized fire activity from slash-and-burn agriculture (Tyukavina et al., 2018). These fires complicate the trends of HCHO and $NO_2$, both due to smoke aerosol interference with satellite retrievals, and transient spikes during extreme events (Jin et al., 2023). Consequently, standard trend analysis cannot reliably resolve emission signals in such fire-prone areas, necessitating specialized methodologies beyond this study's scope.

……

**3.2.2 Drivers of HCHO/$NO_2$ Trend and Trend Reversals**

……

Developed regions exhibited normally linear dominance with subtle shifts, yet displayed distinct multi-phase patterns. In the U.S., initial localized positive trends emerged in the *early 2000s*. These transitions - from flat or weakly increasing baselines to rapid growth - were more evident referring to **Error! Reference source not found.**b. From 1996 to 2015, nationwide HCHO/$NO_2$ ratios increased by 52-124%, primarily driven by $NO_x$ reductions mandated by the 1990 Clean Air Act Amendments (Amendments, 1990). This legislation prompted a strategic shift from VOC-centric controls to integrated $NO_x$-VOC management, directly resulting in a sharp decline in $NO_2$ levels relative to 1996 values by 2015 (~46%) (Figure S3a), while HCHO showed minor fluctuations. The HCHO stability likely stems from persistent biogenic emissions, especially in the Southeast U.S. (De Smedt et al., 2008). Over industrialized areas, anthropogenic emission changes drive HCHO column trends more (Stavrakou et al., 2014; Zhu et al., 2014; De Smedt et al., 2010). Large regional HCHO column variabilities are mainly attributable to fire events and temperature fluctuations (Stavrakou et al., 2014). After 2015, localized reversals appeared, notably in California, where HCHO/$NO_2$ ratios dipped slightly due to plateauing $NO_x$ reductions and marginally declining HCHO (Figue S3). However, these localized reversals did not coalesce into broader regional trends.

Europe exhibits pronounced spatiotemporal heterogeneity in HCHO/$NO_2$ ratio trends compared to the relatively uniform pattern observed across the U.S. The continental-scale increase, initiated in the early 2000s, demonstrates strong $NO_2$ reduction dominance, while temporal fluctuations primarily reflect HCHO variability (Figure S3). This overall upward trajectory aligns with the EU's 1996 Integrated Pollution Prevention and Control Directive, which mandated sector-specific emission standards for refineries and chemical industries. Despite coordinated EU-wide air quality policies, national outcomes vary significantly: The UK achieved an 80% reduction in $NO_x$ emissions since 1990 (from over 3 million tons to

677,500 tons in 2021) (Statista Data), while France cut $NO_x$ by 61% over the past two decades (reaching 651,000 metric tons in 2023), largely through transportation reforms. Spain saw a 55% decline since 1990 (588,100 tons in 2022), likely due to power plant emission controls (Curier et al., 2014). These divergent trends stem from variations in national $NO_x$ sources and policy effectiveness (Jamali et al., 2020; Paraschiv et al., 2017), as well as persistent non-compliance issues—particularly in road transport, which accounted for 94% of EU air quality standard exceedances in 2015 (European Environment Agency, 2015). Under these circumstances, the reversal points also varied across nations: Central European countries (e.g., Poland, Czechia) peaked around 2005, while Western Mediterranean nations (e.g., Spain, Italy) and Germany reached their maxima circa 2020, with France and the Netherlands still maintaining an upward trend (**Error! Reference source not found.**a).

In southeastern Australia, while the overall trend in $HCHO/NO_2$ ratios showed weak decline, a distinct transition from negative to positive trends emerged around 2007. This reversal coincides temporally with policy-driven emission reductions implemented following severe haze events in Sydney during the early 1990s, which prompted federal action targeting vehicular pollution. The 2003 National Clean Air Agreement marked a turning point, achieving a 42% reduction in diesel vehicle smoke emissions within three years (https://www.dcceew.gov.au/). The observed 2007 inflection point reflects the time lag between policy implementation and measurable improvements in air quality.

Globally, linear trends predominantly characterize $HCHO/NO_2$ ratio evolution in developed regions (e.g., Western Europe, the U.S., Japan), resulting from sustained emission reduction policies. In contrast, developing regions (e.g., East China) exhibit nonlinear trajectories with marked reversals, attributable to rapid industrialization coupled with delayed implementation of emission controls. These patterns consist with satellite-derived $NO_2$ trends (Georgoulias et al., 2019) and VOC dynamics studies (Fan et al., 2023; Kuttippurath et al., 2022; De Smedt et al., 2015; De Smedt et al., 2008). These studies confirm that meteorological factors could explain sub-decadal fluctuations, whereas anthropogenic emissions represent the fundamental driver of long-term trends.

3. *The paper could be more concise and better organized. (1) Similar interpretations of trends in $HCHO/NO_2$ and the WE/WD effect for East Asia and US appear repeatedly without deeper insight. These discussions could be condensed and analyzed within a single section.*

Reply: In the revised version, we have consolidated the analyses of $HCHO/NO_2$ trends and weekday-weekend (WE-WD) $O_3$ differences into a unified Section 3.3 to improve clarity and eliminate redundancy. These two metrics provide complementary insights into $O_3$ regime transitions: The WE-WD $O_3$ effect offers direct observational evidence of regime shifts, which are valuable in regions with extensive monitoring records (e.g., Europe, the U.S., and Japan), while satellite-derived $HCHO/NO_2$ ratios provide continuous spatial coverage, enabling analysis in regions with shorter observational histories (e.g., urban China and India). This integrated approach allows assessment across all regions regardless of surface monitoring network maturity. We have revised Section 3.3, which presents a more streamlined yet thorough examination of $O_3$ chemical regime shifts.

*(2) The purpose of Table 1 and Figure 7 is unclear and somehow very confusing. Figure 8, which presents regional time series of HCHO/NO₂, already conveys the information clearly.*

We value the reviewer's comments on Table 1 and Figure 7. Figure 7(now Figure 10 in revised paper) differ from Figure 8 (now Figure 9 in revised paper): Figure 8 shows regional aggregated time series, while Figure 7 shows the spatial variabilities of the evolution of $O_3$ production regimes, which could reveal regions with diverse production regimes. Figure 7 can also identify where and when the regime transition occurs, which represents a key novelty of this study. Table 1 represents another key novelty of this study, which incorporates the long-term changes of $HCHO/NO_2$ into the definition of regimes, which informs both the status and the historical trends of the $O_3$ production regimes. In response to the reviewer's feedback regarding clarity of intent, we have improved the explanations of methods and findings in the revised Section 3.4 ("Global Spatiotemporal Evolution in $O_3$ Chemical Regimes: Transition Status and Potential Transition Years"). The revised version has described in detail the analytical approach's objectives, the classification methodology, and how this complements the time series analysis.

4. *The free-tropospheric (background) contribution to the satellite-observed $NO_2$ columns is increasingly important, especially in the US and Europe, where surface emissions have declined continuously. Could this alter the threshold (should the threshold remain fixed over time?), given the reduced representativeness of the column for surface conditions due to $NO_x$ reduction? Additionally, please consider discussing how background interference might affect the results.*

Reply: That's a good point. We use tropospheric $NO_2$ and HCHO columns to diagnose the near surface $O_3$ chemistry, which is affected by the varying column-to-surface relationships. The column-to-surface relationships are affected by boundary layer dynamics and the contribution from free troposphere. However, the impacts on the regime threshold values should be small over source regions, where the spatial and temporal trends of $NO_2$ are dominated by anthropogenic emissions. For the regions with decreasing $NO_2$, the contribution of free tropospheric $NO_2$ is likely increase (Dang et al., 2023) , which could bias the observed trends of $HCHO/NO_2$, but not the overall direction of the regime changes. In the revised manuscript, we have added discussions about the contribution of free-tropospheric $NO_2$ columns to interpretation of long-term trends of $NO_2$:

However, it should be noted that using satellite $HCHO/NO_2$ to diagnose $O_3$ production regimes is subject to uncertainties of satellite retrievals and the regime threshold values. We use tropospheric $NO_2$ and HCHO column densities to infer near-surface $O_3$ chemistry, but this approach is influenced by variable column-to-surface relationships driven by boundary layer dynamics and contributions from the free troposphere (Jin et al., 2017, Dang et al., 2023, Wolfe et al., 2019). Especially for the regions with decreasing $NO_2$, the contribution of free tropospheric $NO_2$ is likely increase (Dang et al., 2023), which could bias the observed trends of $HCHO/NO_2$.

*Minor comments:*

1. *Section 2.1, this harmonized GOME, SCIAMACHY, OMI, and TROPOMI product should be better described, including how biases across different platforms are addressed.*

Reply: We have added detailed methodologies for the harmonization of the multi-satellite products in the revised manuscript:

To investigate the long-term changes in $HCHO/NO_2$, we construct annual average tropospheric $NO_2$ and HCHO VCD data from the GOME (1996-2001), SCIAMACHY (2002-2003) and OMI (2004-2020) and TROPOMI (2020 - 2022) datasets. GOME and SCIAMACHY and TROPOMI data are harmonized with reference to OMI data with a resolution of $0.25° \times 0.25°$. The retrieval and harmonization scheme are described in Jin et al. (2020). Briefly, we use OMI as a reference to adjust GOME and SCIAMACHY columns as OMI has the finest spatial resolution and the overpass time of interest where captures the most active $O_3$ formation chemistry. For $NO_2$, the difference among satellite instruments is decomposed to two components: (1) difference due to resolution; (2) difference due to overpass time. The difference due to resolution is adjusted by comparing the differences in re-gridding Level-2 OMI $NO_2$ to fine-resolution ($0.25° \times 0.25°$) grid versus a coarse-resolution ($2° \times 0.5°$, resolution closer to that of GOME) grid. The difference in overpass time is derived from the mean difference between OMI and SCIAMACHY during overlapping years (2004 to 2012) at a coarse resolution ($2° \times 0.5°$). For HCHO, as the spatial variations of HCHO are mostly regional, the harmonization only accounts for the difference caused by overpass time (Jin et al., 2020). We grid all Level-2 satellite HCHO products to $0.25° \times 0.25°$, and adjust GOME and SCIAMACHY HCHO columns by adding the mean difference between SCIAMACHY and OMI during the overlapping period. We do not adjust for the difference between OMI and TROPOMI as their overpass time is close.

2. *Line 97-99, what do you mean by "using the same a priori profile"? Does the TM5-MP provide a priori shape factors for all instruments, and does its emission input change annually/monthly?*

Reply: We meant that the calculations of air mass factors for satellite retrievals of multi-instrument HCHO and $NO_2$ uses the same model simulations from TM5-MP that features consistent meteorology, emissions and chemical mechanisms. The details of the model simulations can be found in (Williams et al., 2017) . We have clarified this point in the manuscript as follows:

We use satellite-based products developed under the Quality Assurance for Essential Climate Variables (QA4ECV) project, which retrieves $NO_2$ and HCHO consistently using the same

model simulations from TM5-MP as a priori profile that features consistent meteorology, emissions and chemical mechanisms (Boersma et al., 2018; Boersma et al., 2017b, a; De Smedt et al., 2017; Williams et al., 2017).

3. *Figure 4, the presence of many discrete low values over Inner Mongolia in East Asia seems unreasonable. Could this be an issue with satellite observations, or is there another possible explanation?*

Reply: That's a good point. We also notice the anomalously low HCHO/NO₂ ratios observed over Inner Mongolia (35–42°N, 105–110°E). In the original manuscript, the mean HCHO/NO₂ is calculated from annual HCHO/NO₂, which is affected by anomalously low values in some years. We have revised the method to by first computing multi-year average HCHO and NO₂ concentrations separately, then deriving the HCHO/NO₂ ratio. Using the revised method removes all those noisy and anomalously low HCHO/NO₂ values. We have revised this figure in the revised manuscript:

[Figure]

It should be noted that low ratios in Inner Mongolia can still be found, which is consistent with independent satellite observations (Wang et al. (2021), their Figure 2).This region corresponds to the Hohhot-Baotou-Ordos urban cluster, a major industrial hub in northern China. Key emission sources include: Coal-fired power plants and Heavy industry (Ordos hosts one of China's largest coal-power bases, while Baotou's steel industry requires significant coal combustion for coking. Baotou's steel and rare earth industries account for over 40% of municipal NOₓ emissions (Inner Mongolia Pollution Census Bulletin, 2020, http://sthjt.nmg.gov.cn/). Ordos is a key national coal chemical industry base. Vehicular growth (The combined vehicle fleet across the three cities exceeds 3 million, with annual growth rates of 9–12% since 2012). In 2019 alone, Baotou's industrial NOₓ emissions reached 126,000 metric tons (23% of Inner Mongolia's total; MEE's 2+26 Cities Monitoring Report, https://www.mee.gov.cn/). These emission intensities create a persistent high-NOₓ

environment, suppressing satellite HCHO/NO₂ ratios. Satellite observations thus accurately reflect real-world conditions, consistent with ground inventories.

*4. Line 23, "WHO" should be capitalized.*

Reply: Done.

*5. Line 275, there is no label of "a" in Figure S3.*

Reply: The missing panel label "(a)" has been added.

*6. Line 285-286, please provide references to support this statement.*

Reply: We have added new references in the revised manuscript as follows:

This legislation prompted a strategic shift from VOC-centric controls to integrated $NO_x$-VOC management, directly resulting in a significant reduction in $NO_2$ levels across the U.S. since 2000 (Duncan et al., 2016; Jin et al., 2020; Lamsal et al., 2015) , but the trends of HCHO are flat, largely due to the contributions from biogenic VOCs (Jin et al., 2020; Zhu et al., 2017).

**Reference:**

Adams, T. J., Geddes, J. A., and Lind, E. S.: New Insights Into the Role of Atmospheric Transport and Mixing on Column and Surface Concentrations of NO2 at a Coastal Urban Site, Journal of Geophysical Research: Atmospheres, 128, 10.1029/2022jd038237, 2023.

Amendments, C. A. A.: Clean Air Act Amendments of 1990, Pub. L. No. 101-549, 104 Stat. 2399, 1990.

Andela, N., Morton, D. C., Giglio, L., Chen, Y., van der Werf, G. R., Kasibhatla, P. S., DeFries, R. S., Collatz, G. J., Hantson, S., Kloster, S., Bachelet, D., Forrest, M., Lasslop, G., Li, F., Mangeon, S., Melton, J. R., Yue, C., and Randerson, J. T.: A human-driven decline in global burned area, Science, 356, 1356-1362, 10.1126/science.aal4108, 2017.

Boersma, K. F., Eskes, H., Richter, A., De Smedt, I., Lorente, A., Beirle, S., Van Geffen, J., Peters, E., Van Roozendael, M., and Wagner, T.: QA4ECV NO2 tropospheric and stratospheric column data from OMI, Royal Netherlands Meteorological Institute (KNMI), 10.21944/qa4ecv-no2-omi-v1.1, 2017a.

Boersma, K. F., Eskes, H., Richter, A., De Smedt, I., Lorente, A., Beirle, S., Van Geffen, J., Peters, E., Van Roozendael, M., and Wagner, T.: QA4ECV NO2 tropospheric and stratospheric column data from GOME, Royal Netherlands Meteorological Institute (KNMI), 10.21944/qa4ecv-no2-gome-v1.1, 2017b.

Boersma, K. F., Eskes, H. J., Richter, A., De Smedt, I., Lorente, A., Beirle, S., van Geffen, J. H. G. M., Zara, M., Peters, E., Van Roozendael, M., Wagner, T., Maasakkers, J. D., van der A, R. J., Nightingale, J., De Rudder, A., Irie, H., Pinardi, G., Lambert, J.-C., and Compernolle, S. C.: Improving algorithms and uncertainty estimates for satellite NO2 retrievals: results from the quality assurance for the essential climate variables (QA4ECV) project, Atmospheric

Measurement Techniques, 11, 6651-6678, 10.5194/amt-11-6651-2018, 2018.

Chan Miller, C., Jacob, D. J., González Abad, G., and Chance, K.: Hotspot of glyoxal over the Pearl River delta seen from the OMI satellite instrument: implications for emissions of aromatic hydrocarbons, Atmospheric Chemistry and Physics, 16, 4631-4639, 10.5194/acp-16-4631-2016, 2016.

Curier, R. L., Kranenburg, R., Segers, A. J. S., Timmermans, R. M. A., and Schaap, M.: Synergistic use of OMI NO2 tropospheric columns and LOTOS–EUROS to evaluate the NOx emission trends across Europe, Remote Sensing of Environment, 149, 58-69, 10.1016/j.rse.2014.03.032, 2014.

De Smedt, I., Stavrakou, T., Müller, J. F., van der A, R. J., and Van Roozendael, M.: Trend detection in satellite observations of formaldehyde tropospheric columns, Geophysical Research Letters, 37, 10.1029/2010gl044245, 2010.

De Smedt, I., Müller, J. F., Stavrakou, T., van der A, R., Eskes, H., and Van Roozendael, M.: Twelve years of global observations of formaldehyde in the troposphere using GOME and SCIAMACHY sensors, Atmospheric Chemistry and Physics, 8, 4947-4963, 10.5194/acp-8-4947-2008, 2008.

De Smedt, I., YU, H., Richter, A., Beirle, S., Eskes, H., Boersma, K. F., Van Roozendael, M., Van Geffen, J., Wagner, T., Lorente, A., and Peters, E.: QA4ECV HCHO tropospheric column data from OMI, Royal Belgian Institute for Space Aeronomy, 10.18758/71021031, 2017.

De Smedt, I., Stavrakou, T., Hendrick, F., Danckaert, T., Vlemmix, T., Pinardi, G., Theys, N., Lerot, C., Gielen, C., Vigouroux, C., Hermans, C., Fayt, C., Veefkind, P., Müller, J. F., and Van Roozendael, M.: Diurnal, seasonal and long-term variations of global formaldehyde columns inferred from combined OMI and GOME-2 observations, Atmospheric Chemistry and Physics, 15, 12519-12545, 10.5194/acp-15-12519-2015, 2015.

Fan, J., Wang, T., Wang, Q., Ma, D., Li, Y., Zhou, M., and Wang, T.: Assessment of HCHO in Beijing during 2009 to 2020 using satellite observation and numerical model: Spatial characteristic and impact factor, Science of the Total Environment, 894, 165060-165072, 10.1016/j.scitotenv.2023.165060, 2023.

Georgoulias, A. K., van der A, R. J., Stammes, P., Boersma, K. F., and Eskes, H. J.: Trends and trend reversal detection in 2 decades of tropospheric NO2 satellite observations, Atmospheric Chemistry and Physics, 19, 6269-6294, 10.5194/acp-19-6269-2019, 2019.

Hilboll, A., Richter, A., and Burrows, J. P.: Long-term changes of tropospheric NO2 over megacities derived from multiple satellite instruments, Atmospheric Chemistry and Physics, 13, 4145-4169, 10.5194/acp-13-4145-2013, 2013.

Jamali, S., Klingmyr, D., and Tagesson, T.: Global-Scale Patterns and Trends in Tropospheric NO2 Concentrations, 2005–2018, Remote Sensing, 12, 10.3390/rs12213526, 2020.

Jin, X., Fiore, A. M., and Cohen, R. C.: Space-Based Observations of Ozone Precursors within California Wildfire Plumes and the Impacts on Ozone-NO(x)-VOC Chemistry, Environ Sci Technol, 57, 14648-14660, 10.1021/acs.est.3c04411, 2023.

Jin, X., Fiore, A., Boersma, K. F., Smedt, I., and Valin, L.: Inferring changes in summertime surface ozone-NOx-VOC chemistry over U.S. urban areas from two decades of satellite and ground-based observations, Environmental Science & Technology, 54, 6518-6529, 10.1021/acs.est.9b07785, 2020.

Kuttippurath, J., Abbhishek, K., Gopikrishnan, G. S., and Pathak, M.: Investigation of long–term trends and major sources of atmospheric HCHO over India, Environmental Challenges, 7, 10.1016/j.envc.2022.100477, 2022.

Li, K., Jacob, D. J., Liao, H., Shen, L., Zhang, Q., and Bates, K. H.: Anthropogenic drivers of 2013-2017 trends in summer surface ozone in China, Proc Natl Acad Sci U S A, 116, 422-427, 10.1073/pnas.1812168116, 2019.

Liu, Z., Wang, Y., Gu, D., Zhao, C., Huey, L. G., Stickel, R., Liao, J., Shao, M., Zhu, T., Zeng, L., Amoroso, A., Costabile, F., Chang, C. C., and Liu, S. C.: Summertime photochemistry during CAREBeijing-2007: ROx budgets and O3 formation, Atmospheric Chemistry and Physics, 12, 7737-7752, 10.5194/acp-12-7737-2012, 2012.

Palmer, P. I., Barkley, M. P., Kurosu, T. P., Lewis, A. C., Saxton, J. E., Chance, K., and Gatti, L. V.: Interpreting satellite column observations of formaldehyde over tropical South America, Philos Trans A Math Phys Eng Sci, 365, 1741-1751, 10.1098/rsta.2007.2042, 2007.

Palmer, P. I., Abbot, D. S., Fu, T. M., Jacob, D. J., Chance, K., Kurosu, T. P., Guenther, A., Wiedinmyer, C., Stanton, J. C., Pilling, M. J., Pressley, S. N., Lamb, B., and Sumner, A. L.: Quantifying the seasonal and interannual variability of North American isoprene emissions using satellite observations of the formaldehyde column, Journal of Geophysical Research: Atmospheres, 111, 10.1029/2005jd006689, 2006.

Paraschiv, S., Constantin, D. E., Paraschiv, S. L., and Voiculescu, M.: OMI and Ground-Based In-Situ Tropospheric Nitrogen Dioxide Observations over Several Important European Cities during 2005-2014, Int J Environ Res Public Health, 14, 10.3390/ijerph14111415, 2017.

Shen, L., Jacob, D. J., Zhu, L., Zhang, Q., Zheng, B., Sulprizio, M. P., Li, K., De Smedt, I., González Abad, G., Cao, H., Fu, T. M., and Liao, H.: The 2005–2016 Trends of Formaldehyde Columns Over China Observed by Satellites: Increasing Anthropogenic Emissions of Volatile Organic Compounds and Decreasing Agricultural Fire Emissions, Geophysical Research Letters, 46, 4468-4475, 10.1029/2019gl082172, 2019.

Stavrakou, T., Müller, J. F., Bauwens, M., De Smedt, I., Van Roozendael, M., Guenther, A., Wild, M., and Xia, X.: Isoprene emissions over Asia 1979–2012: impact of climate and land-use changes, Atmospheric Chemistry and Physics, 14, 4587-4605, 10.5194/acp-14-4587-2014, 2014.

Tyukavina, A., Hansen, M. C., Potapov, P., Parker, D., Okpa, C., Stehman, S. V., Kommareddy, I., and Turubanova, S.: Congo Basin forest loss dominated by increasing smallholder clearing, Sci Adv, 4, eaat2993, 10.1126/sciadv.aat2993, 2018.

Wang, W., van der A, R., Ding, J., van Weele, M., and Cheng, T.: Spatial and temporal changes of the ozone sensitivity in China based on satellite and ground-based observations, Atmospheric Chemistry and Physics, 21, 7253-7269, 10.5194/acp-21-7253-2021, 2021.

Wells, K. C., Millet, D. B., Payne, V. H., Deventer, M. J., Bates, K. H., de Gouw, J. A., Graus, M., Warneke, C., Wisthaler, A., and Fuentes, J. D.: Satellite isoprene retrievals constrain emissions and atmospheric oxidation, Nature, 585, 225-233, 10.1038/s41586-020-2664-3, 2020.

Williams, J. E., Boersma, K. F., Le Sager, P., and Verstraeten, W. W.: The high-resolution version of TM5-MP for optimized satellite retrievals: description and validation, Geoscientific Model Development, 10, 721-750, 10.5194/gmd-10-721-2017, 2017.

Zhang, Y., Cooper, O. R., Gaudel, A., Nedelec, P., Ogino, S. Y., Thompson, A. M., and West, J. J.: Tropospheric ozone change from 1980 to 2010 dominated by equatorward redistribution of emissions, Nat Geosci, 9, 875-879, 10.1038/NGEO2827, 2016a.

Zhang, Y., Wang, Y., Chen, G., Smeltzer, C., Crawford, J., Olson, J., Szykman, J., Weinheimer, A. J., Knapp, D. J., Montzka, D. D., Wisthaler, A., Mikoviny, T., Fried, A., and Diskin, G.: Large vertical gradient of reactive nitrogen oxides in the boundary layer: Modeling analysis of DISCOVER-AQ 2011 observations, Journal of Geophysical Research: Atmospheres, 121, 1922-1934, 10.1002/2015jd024203, 2016b.

Zhu, L., Jacob, D. J., Mickley, L. J., Marais, E. A., Cohan, D. S., Yoshida, Y., Duncan, B. N., González Abad, G., and Chance, K. V.: Anthropogenic emissions of highly reactive volatile organic compounds in eastern Texas inferred from oversampling of satellite (OMI) measurements of HCHO columns, Environmental Research Letters, 9, 10.1088/1748-9326/9/11/114004, 2014.

---

## Author Comment (AC2)

**Reply to Reviewer 2**

*The article presents a novel approach to inferring ozone formation sensitivity on a global scale. The authors combine two widely used indicators—the ozone weekend effect (WE-WD $O_3$) and the formaldehyde-to-nitrogen dioxide ratio ($HCHO/NO_2$)—to determine regime thresholds. By correlating these variables and applying linear regression, the $HCHO/NO_2$ threshold for regime transition is identified as the point where WE-WD $O_3$ shifts from positive to negative values. The study includes an extensive trend analysis and trend reversal evaluation, ultimately establishing a global threshold of 3.5, with regional variations. This approach makes a valuable contribution to ozone mitigation strategies by providing a framework for more precise regime classification on a global scale. I suggest several revisions and clarifications before the paper can be considered for publication in Atmospheric Chemistry and Physics.*

Reply: We would like to thank the reviewers for their time to review this paper. We have revised the manuscript following the reviewers' suggestions. Below are our point-by-point responses to the comments, along with the corresponding revisions.

***Specific Comments:***

1. *Given the nonlinear nature of $O_3$ formation due to complex VOC-$NO_x$ interactions, comparing linear and nonlinear models would help justify the choice of linear regression for deriving regime threshold values.*

Reply: We acknowledge that $O_3$ formation is nonlinearly dependent on $NO_x$ and VOCs. The $O_3$ weekend effect captures this nonlinear dependence on $NO_x$, as it reflects the sensitivity of $O_3$ to changes in $NO_x$ emissions on weekends, i.e., the derivative of $O_3$ with respect to $NO_x$. The transitioning point at which WE-WD = 0 is effectively at which $d[O_3]/d[NO_x] = 0$, which represent the transitioning point at which $O_3$ sensitivity to $NO_x$ emission changes signs. Therefore, we think the linear regression method should be sufficient to capture the nonlinear $O_3$-$NO_x$-VOC chemistry.

We also tested polynomial models of increasing complexity to examine the relationship between $HCHO/NO_2$ ratios and WE-WD $O_3$ differences. Our analysis shows that while higher-order polynomial models (up to cubic terms) do offer slightly improved statistical performance - as evidenced by reduced RMSE and increased $R^2$ values (see Figures 1-3 below) - these improvements are relatively modest in magnitude. More importantly, the nonlinear terms in these more complex models currently lack clear interpretation within the framework of $O_3$ formation regimes. After carefully weighing both the statistical and physicochemical considerations, we concluded that the original linear regression approach remains preferable. It provides sufficient fitting accuracy and straightforward physical interpretability, which consistent with the $O_3$ chemistry. We believe this balanced approach

ensures our threshold determination remains both statistically sound and chemically meaningful.

In the revised manuscript, we explain the reason why WE-WD can effectively capture the nonlinear $O_3$-$NO_x$-VOC chemistry as follows:

The WE-WD $O_3$ difference reflects the sensitivity of $O_3$ to emission reduction in $NO_x$ on weekends, which is effectively the derivative of $O_3$ with respect to $NO_x$, and the transitioning point at which $O_3$ weekend effect crosses zero represents the transitioning point at which $O_3$ sensitivity to $NO_x$ emission changes signs, which often corresponds to the peak $O_3$ production.

We derive threshold values for the HCHO/$NO_2$ ratio that delineate $O_3$ formation regimes by correlating the WE–WD differences in $O_3$ with HCHO/$NO_2$ using linear regression. The regime threshold corresponds to the intercept (zero-crossing point) of the regression line, where the sign of WE–WD $O_3$ changes.

[Figure]

Figure 1: Scatter plot of the monthly average satellite-derived HCHO/$NO_2$ ratio versus the WE-WD $O_3$ concentration in 9 representative cities. The black line shows the fitted linear regression line with red triangles indicating inflection points where the regression line intersects the WE-WD $O_3$ = 0 baseline.

[Figure]

Figure 2: Second-order polynomial regression.

[Figure]

Figure 3: Third-order polynomial regression.

2. *A gradual transition between VOC-limited and NOₓ-limited regimes is expected rather than an abrupt shift at a single point. Could nonlinear regression methods better capture this transition?*

Reply: We agree that the transition between VOC-limited and NOₓ-limited regimes should be gradual rather than an abrupt shift. In our original manuscript, we employed a fixed threshold value primarily for simplification purposes, while recognizing this approach may not fully capture the transitional nature of regime shifts. The regime thresholds have uncertainties, and previous studies typically assume a range for regime threshold values (Jin et al., 2020; Jin et al., 2017; Sillman, 1999). To address this important point, we have improved our methodology in the revised manuscript:

1.  Enhanced threshold characterization: When aggregating regional site data, we now not only identify the most frequent HCHO/NO₂ thresholds (peak HCHO/NO₂ frequencies, ($\tilde{x}$)) but also determine the transitional range [$x_{lower}$-$x_{upper}$] based on the top 10% frequency interval to better quantify uncertainty, as illustrated in the new Figures 3b-c below.
2.  For regional-scale analysis (Figure 9), we now employ region specific threshold ranges derived from Figure 3c. For grid-level assessment (Figure 10), we utilize regionally aggregated threshold ranges identified in Figure 3b, rather than using a global uniform value.

These improvements provide more accurate identification of O₃ regime transitions across different geographical contexts. We have accordingly updated all relevant discussions in Sections 3.3 and 3.4.

[Figure]

Figure 3: (b) Frequency distribution of HCHO/NO₂ thresholds across regions (R² > 0.2). Solid lines denote continents with >1500 valid sites; dashed lines represent regions with <150 sites. (c) Box plots of transition thresholds in economically advanced regions (marked in (a)). Black numbers: peak HCHO/NO₂ frequencies ($\tilde{x}$); red range: top 10% frequency interval defining the transitional range [$x_{lower}$, $x_{upper}$].

3. *The long-term trend and trend reversal analyses rely on datasets with different spatial and temporal resolutions and measurement times. Although the datasets were harmonized, these differences may introduce biases in the observed trends and reversals. Were any considerations made to assess these biases?*

Reply: We have accounted for the differences in resolution and overpass time for the harmonization of satellite data products. Briefly, we use OMI as a reference to adjust GOME and SCIAMACHY columns as OMI has the finest spatial resolution and the overpass time of interest where captures the most active $O_3$ formation chemistry. For $NO_2$, the difference among satellite instruments is decomposed to two components: (1) differences caused by resolution; (2) difference due to overpass time. The difference caused by resolution is adjusted by comparing the differences in re-gridding Level-2 OMI $NO_2$ to fine-resolution (0.25° × 0.25°) grid versus a coarse-resolution (2° × 0.5°, resolution closer to that of GOME) grid. The difference in overpass time is derived from the mean difference between OMI and SCIAMACHY during overlapping years (2004 to 2012) at a coarse resolution (2° × 0.5°). For HCHO, as the spatial variations of HCHO are mostly regional, the harmonization only accounts for the difference caused by overpass time (Jin et al., 2020). We grid all Level-2 satellite HCHO products to 0.25° × 0.25°, and we adjust GOME and SCIAMACHY HCHO columns by adding the mean difference between SCIAMACHY and OMI during the overlapping period. We do not adjust for the difference between OMI and TROPOMI as their overpass time is close.

We have added detailed methodologies for the harmonization of the multi-satellite products in the revised manuscript:

To investigate the long-term changes in HCHO/$NO_2$, we construct annual average tropospheric $NO_2$ and HCHO VCD data from the GOME (1996-2001), SCIAMACHY (2002-2003) and OMI (2004-2020) and TROPOMI (2020 - 2022) datasets. GOME and SCIAMACHY and TROPOMI data are harmonized with reference to OMI data with a resolution of 0.25˚ × 0.25˚. The retrieval and harmonization scheme are described in Jin et al. (2020). Briefly, we use OMI as a reference to adjust GOME and SCIAMACHY columns as OMI has the finest spatial resolution and the overpass time of interest where captures the most active $O_3$ formation chemistry. For $NO_2$, the difference among satellite instruments is decomposed to two components: (1) difference due to resolution; (2) difference due to overpass time. The difference due to resolution is adjusted by comparing the differences in re-gridding Level-2 OMI $NO_2$ to fine-resolution (0.25° × 0.25°) grid versus a coarse-resolution (2° × 0.5°, resolution closer to that of GOME) grid. The difference in overpass time is derived from the mean difference between OMI and SCIAMACHY during overlapping years (2004 to 2012) at a coarse resolution (2° × 0.5°). For HCHO, as the spatial variations of HCHO are mostly regional, the harmonization only accounts for the difference

caused by overpass time (Jin et al., 2020). We grid all Level-2 satellite HCHO products to $0.25° \times 0.25°$, and adjust GOME and SCIAMACHY HCHO columns by adding the mean difference between SCIAMACHY and OMI during the overlapping period. We do not adjust for the difference between OMI and TROPOMI as their overpass time is close.

4. *The method used for trend and trend reversal evaluations assumes that trends are linear within 5-year windows and does not account for seasonality. Complementary statistical methods, such as seasonal-trend decomposition, could help verify the detected trends and reversals.*

Reply: Thanks for the reviewer's suggestion to address seasonality in our trend analysis. We used annual averages here for two reasons. First, satellite products (especially GOME and SCIAMACHY) at monthly scale have large uncertainties and noise, which would affect the derived trend. Second, while HCHO and $NO_2$ show seasonal variations, these cyclical patterns primarily manifest as short-term fluctuations that don't fundamentally alter the long-term decadal trends - similar to the $NO_x$ trends shown in Georgoulias et al. (2019).

Our 5-year moving window approach was chosen to prioritize detection of persistent, policy-driven emission changes over sub-annual variability. This aligns with our study's focus on decadal-scale $O_3$ regime evolution rather than seasonal-scale variations.

We fully agree that seasonal-trend decomposition could provide valuable additional insights, particularly for identifying phase shifts (e.g., intensifying winter $NO_x$ saturation in transitioning economies). While such analysis is beyond our current scope, we've added this as an important future research direction in the conclusion section:

Here, we examine the $O_3$ regime changes based on annual average HCHO/$NO_2$, but the $O_3$ chemical regime should vary seasonally (Jin et al., 2017; Jacob et al., 1995), typically becoming more $NO_x$-saturated in wintertime and more $NO_x$-limited in summertime. We exclude seasonal analysis because varying climatic definitions across regions would complicate cross-regional comparisons, and these cyclical variations do not substantially affect long-term decadal trends.

**Conclusions:** Here we focus on $O_3$ regime evolution annually, but $O_3$ regime also varies seasonally and diurnally. How the seasonal and diurnal variations of $O_3$ regime have evolved over time warrants further investigation. Further research could employ chemical-transport modeling to better understand both seasonal influences and the physical drivers of regional threshold differences, for instance, examining why economically developed regions characterized by higher values compared to less industrialized areas.

5. *Providing more details on the methodology for linking satellite and ground-based observations would improve clarity. Specifically, was the monthly WE-WD $O_3$ value calculated using all available hourly $O_3$ observations? Additionally, were all ground monitoring stations within a 0.25° grid included in computing the WE-WD $O_3$ averages before pairing with HCHO/$NO_2$ data?*

Reply: In the revised manuscript, we have added details in the description of the satellite-ground data linking method. Sundays were selected to represent weekends, and Tuesdays to Thursdays (mid-week days) to represent weekdays. The WE-WD $O_3$ analysis serves two parts of our study. One is in threshold determination (Figures 2-3): To establish robust relationships, we paired ground-based WE-WD $O_3$ data with the nearest grid HCHO/NO2 ratio data. The other is in regional trend analysis (Figures 7-8). Here, we included all ground stations within a region. This strategy balances precision (for threshold derivation) and representativeness (for regional trends), ensuring methodological rigor while addressing distinct analytical objectives. The methodology for linking WE-WD with satellite HCHO/$NO_2$ is added in the revised manuscript:

The WE-WD $O_3$ difference reflects the sensitivity of $O_3$ to emission reduction in $NO_x$ on weekends, which is effectively the derivative of $O_3$ with respect to $NO_x$, and the transitioning point at which $O_3$ weekend effect crosses zero represents the transitioning point at which $O_3$ sensitivity to $NO_x$ emission changes signs, which often corresponds to the peak $O_3$ production. In this study, WE-WD $O_3$ difference is quantified using a standardized protocol: **Sundays** is designated as weekends, while **Tuesdays–Thursdays** is designated weekdays, excluding Mondays and Fridays to minimize transitional effects from adjacent days. For each site and weekly interval throughout the observation period, we calculate the mean differences in WE-WD $O_3$. To calculate long-term trends of WE-WD $O_3$ in Section 3.3, all sites within the region are included. Given the global scope of this analysis and the inherent complexity in defining distinct $O_3$ seasons across various regions, we utilize all-year data without seasonal selection. Using $t$-test at each site to ascertain the statistical significance of WE-WD difference (p-value<0.05). Statistically significant WE-WD differences are identified at each site, and trends were evaluated using 5-year rolling intervals to dampen interannual meteorological variability (Pierce et al., 2010).

To build the relationship between observed $O_3$ weekend effect and satellite HCHO/$NO_2$ (Section 3.1), we mainly use OMI retrievals of HCHO and $NO_2$. OMI is selected as the primary satellite data source due to its unique combination of long-term continuity (2004-2020) and optimal afternoon overpass time. The early afternoon measurement period (13:00-14:00 local time) coincides with peak photochemical activity when $O_3$ production is most active, boundary layer heights are maximized, and solar zenith angles are minimized - all critical factors for obtaining high-quality retrievals of tropospheric HCHO and $NO_2$ columns (Jin et al., 2017; Jin and Holloway, 2015). We derive threshold values for the HCHO/$NO_2$ ratio that delineate $O_3$ formation regimes by correlating the WE–WD differences in $O_3$ with HCHO/$NO_2$ using linear regression. The regime threshold corresponds to the intercept (zero-crossing point) of the regression line, where the sign of WE–WD $O_3$ changes. To establish the relationship between HCHO/$NO_2$ ratios and WE-WD $O_3$, we extract the nearest gridded daily OMI data ($0.125°×0.125°$) corresponding to the ground-based $O_3$ monitoring stations. To ensure precise spatiotemporal matching, we pair the satellite overpass observations with surface measurements by averaging hourly $O_3$ concentrations at 13:00 and 14:00 local time (corresponding to OMI's overpass window).

6. *Although Figure 2 covers 2004–2022, some sites (e.g., Las Vegas, Los Angeles) have a higher number of data points. What explains these differences? Additionally, including the R² value would help assess the regression fit. Clarifying the criteria for selecting the nine representative urban areas would also be recommended.*

Reply: The variation in data point counts across sites arises from differences in observational timelines and data availability within the TOAR-II database. For instance, the Manchester station (the U.K.) operated from 2006–2019 with intermittent gaps during 2007–2010, while Los Angeles (the U.S.) provided continuous measurements from 2004–2022, and Tokyo (Japan) spanned 2012–2021. These disparities are from station-specific factors such as maintenance schedules, data quality control exclusions, and monitoring network development. To ensure robustness, we used monthly averaged WE-WD $O_3$ values and excluded months with fewer than three weeks of valid data, minimizing noise from short-term fluctuations. This may also affect numbers of data points.

The nine representative urban areas in Figure 2 were selected based on two criteria: long-term observational coverage (>10 years) and global representativeness. Sites in regions with sparse long-term data (e.g., South/Southeast Asia, Africa) or non-urban locations were excluded, as their limited records or atypical emission profiles could obscure regime transition signals.

We have supplemented the reasons for such choices in the revised manuscript and included R² values in Figure 2 to quantify regression fit quality:

To demonstrate our approach, we selected nine representative urban stations with long-term (>10 year) records, as shown in Error! Reference source not found.. These sites were chosen ensuring a balanced global representation while factoring in region site density (3 European, 2 North American, 2 Asian, 1 Australian, and 1 Latin American).

7. *Line 205. A discussion on the reasons behind the large spatial variability of threshold values would strengthen the analysis.*

Reply: We have added discussions about the reasons behind the spatial variabilities of the threshold values as follows:

The variations of the regime threshold values of $HCHO/NO_2$ are likely caused by several factors. First, here we use tropospheric column $HCHO/NO_2$ to represent the near-surface $O_3$ chemistry, which is affected by the relationships between column and surface HCHO and $NO_2$ (Jin et al., 2017). The column-to-surface relationship is determined by the boundary layer height and the vertical profiles of HCHO and $NO_2$, which should vary spatially (Adams et al., 2023; Zhang et al., 2016b). Second, HCHO is used as an indicator of VOCs, but the yield of HCHO from oxidation of VOCs varies with different species (Shen et al., 2019;

Chan Miller et al., 2016; Zhu et al., 2014). Regions dominated by biogenic VOC emissions like southeast U.S., tropical regions generally have larger HCHO yield (Wells et al., 2020; Palmer et al., 2007; Palmer et al., 2006). Third, the local chemical environmental may also differ spatially. For example, the lower thresholds in China are consistent with elevated regional $NO_x$ levels (Jamali et al., 2020) and enhanced secondary aerosol formation in this region, which may promote radical loss (Li et al., 2019; Liu et al., 2012). Here we use statistical methods to derive the regime thresholds. Further attribution of the spatial variations is beyond the scope of this study, which warrant further investigation.

8. *Line 247. The methods indicate that trends were calculated using the Mann-Kendall test and Theil-Sen estimator, not linear regression.*

Reply: We thank the reviewer for catching this inconsistency. To clarify: our methodology employs two distinct trend analysis approaches tailored to the data characteristics. For ground-based WE-WD $O_3$ differences (TOAR data), we applied the Theil-Sen estimator with Mann-Kendall significance testing (Figure S5), as surface observations frequently contain localized anomalies (e.g. episodic pollution events) and data gaps across heterogeneous monitoring networks. This non-parametric approach ensures robustness against non-normal distributions and outliers. Conversely, for satellite-derived HCHO/NO₂ ratios, we applied linear regression (Figure 5), as these datasets undergo rigorous preprocessing (cloud filtering, spatial averaging, and anomaly correction) that minimizes outliers and satisfies normality assumptions required for parametric methods.

This differential approach aligns with established practices in atmospheric studies, where Theil-Sen is preferred for heterogeneous surface networks while linear regression remains valid for processed satellite products (Simon et al., 2024; Georgoulias et al., 2019). We have added more clear annotations in Figures 5 and S5 explicitly stating the trend analysis methods:

[Figure]

**Figure 1: Satellite-based linear trends of tropospheric HCHO₂ ratios (1996–2022) for grids with a mean NO₂ VCD > $1.5 \times 10^{15}$ (molecules · cm⁻²) and statistically significant trends at the 95 % confidence level.**

[Figure]

Figure S 1: (a) Map of WE-WD O$_3$ Theil–Sen slope (Raj and Koerts, 1992; Sen, 1968) and (b) p-value for the trend of timeseries using Mann– Kendall test (Kendall, 1975; Mann., 1945).

9.  Line 277. The decline in HCHO/NO$_2$ is attributed to NO$_x$ reductions, but this interpretation appears counterintuitive.

Reply: Thanks for catching this. We have corrected this in the revised manuscript as follows:

Post-2011, industrial VOC emissions rose by 20.46% from 2011-2017 (11,122.7 to 13,397.9 thousand tons/yr; Liu et al. (2021)), amplifying the post-2011 ratio recovery.

10. Line 383. The identification of a single transition year for ozone sensitivity regimes seems somewhat ambiguous, given the temporal and spatial variability of regime classification thresholds, which may have also evolved over time. Focusing on trend changes rather than a specific transition year would provide a more robust interpretation.

Reply: We fully acknowledge that regime classification thresholds may exhibit both temporal and spatial variability, making the determination of a single transition year potentially ambiguous. We consider this analysis valuable not only for determining transition years, but also for establishing a methodological approach to evaluated $O_3$ regime transitions. While previous studies have examined various aspects of $O_3$ regimes, our approach attempts to provide a more systematic quantification of these changes on a grid-level, particularly regarding transition timing and patterns. To address this, we have improved the methods in the revised manuscript by using region-specific transitional ranges defined by the top 10% frequency interval $[x_{lower}\text{-}x_{upper}]$ (Figures 9-10 in the revised paper), and revised the relevant analysis focusing more on identifying trends of transit direction rather than precise years in the new Section 3.4.

**Reference:**

Adams, T. J., Geddes, J. A., and Lind, E. S.: New Insights Into the Role of Atmospheric Transport and Mixing on Column and Surface Concentrations of NO2 at a Coastal Urban Site, Journal of Geophysical Research: Atmospheres, 128, 10.1029/2022jd038237, 2023.

Chan Miller, C., Jacob, D. J., González Abad, G., and Chance, K.: Hotspot of glyoxal over the Pearl River delta seen from the OMI satellite instrument: implications for emissions of aromatic hydrocarbons, Atmospheric Chemistry and Physics, 16, 4631-4639, 10.5194/acp-16-4631-2016, 2016.

Georgoulias, A. K., van der A, R. J., Stammes, P., Boersma, K. F., and Eskes, H. J.: Trends and trend reversal detection in 2 decades of tropospheric NO2 satellite observations, Atmospheric Chemistry and Physics, 19, 6269-6294, 10.5194/acp-19-6269-2019, 2019.

Jacob, D. J., Horowitz, L. W., Munger, J. W., Heikes, B. G., Dickerson, R. R., Artz, R. S., and Keene, W. C.: Seasonal transition from NOx- to hydrocarbon-limited conditions for ozone production over the eastern United States in September, Journal of Geophysical Research, 100, 9315-9324, 10.1029/94jd03125, 1995.

Jamali, S., Klingmyr, D., and Tagesson, T.: Global-Scale Patterns and Trends in Tropospheric NO2 Concentrations, 2005–2018, Remote Sensing, 12, 10.3390/rs12213526, 2020.

Jin, X. and Holloway, T.: Spatial and temporal variability of ozone sensitivity over China observed from the Ozone Monitoring Instrument, Journal of Geophysical Research: Atmospheres, 120, 7229-7246, 10.1002/2015jd023250, 2015.

Jin, X., Fiore, A., Boersma, K. F., Smedt, I., and Valin, L.: Inferring changes in summertime surface ozone-NOx-VOC chemistry over U.S. urban areas from two decades of satellite and ground-based observations, Environmental Science & Technology, 54, 6518-6529, 10.1021/acs.est.9b07785, 2020.

Jin, X., Fiore, A. M., Murray, L. T., Valin, L. C., Lamsal, L. N., Duncan, B., Boersma, K. F., De Smedt, I., Abad, G. G., Chance, K., and Tonnesen, G. S.: Evaluating a space-based indicator of surface ozone-NOx-VOC sensitivity over midlatitude source regions and application to decadal trends, Journal of Geophysical Research: Atmospheres, 122, 10439-10488, 10.1002/2017JD026720, 2017.

Kendall, M. G.: Rank Correlation Methods, 4th edn., Charles Grif- fin, London1975.

Li, K., Jacob, D. J., Liao, H., Shen, L., Zhang, Q., and Bates, K. H.: Anthropogenic drivers of 2013-2017 trends in summer surface ozone in China, Proc Natl Acad Sci U S A, 116, 422-427, 10.1073/pnas.1812168116, 2019.

Liu, R., Zhong, M., Zhao, X., Lu, S., Tian, J., Li, Y., Hou, M., Liang, X., Huang, H., Fan, L., and

Ye, D.: Characteristics of industrial volatile organic compounds(VOCs) emission in China from 2011 to 2019, Environmental Science, 42, 5169-5179, 2021.

Liu, Z., Wang, Y., Gu, D., Zhao, C., Huey, L. G., Stickel, R., Liao, J., Shao, M., Zhu, T., Zeng, L., Amoroso, A., Costabile, F., Chang, C. C., and Liu, S. C.: Summertime photochemistry during CAREBeijing-2007: ROx budgets and O3 formation, Atmospheric Chemistry and Physics, 12, 7737-7752, 10.5194/acp-12-7737-2012, 2012.

Mann., H. B.: Nonparametric Tests Against Trend, Econometrica, 13, 245–259, 10.2307/1907187 1945.

Palmer, P. I., Barkley, M. P., Kurosu, T. P., Lewis, A. C., Saxton, J. E., Chance, K., and Gatti, L. V.: Interpreting satellite column observations of formaldehyde over tropical South America, Philos Trans A Math Phys Eng Sci, 365, 1741-1751, 10.1098/rsta.2007.2042, 2007.

Palmer, P. I., Abbot, D. S., Fu, T. M., Jacob, D. J., Chance, K., Kurosu, T. P., Guenther, A., Wiedinmyer, C., Stanton, J. C., Pilling, M. J., Pressley, S. N., Lamb, B., and Sumner, A. L.: Quantifying the seasonal and interannual variability of North American isoprene emissions using satellite observations of the formaldehyde column, Journal of Geophysical Research: Atmospheres, 111, 10.1029/2005jd006689, 2006.

Raj, B. and Koerts, J.: Henri Theil's Contributions to Economics and Econometrics, Advanced Studies in Theoretical and Applied Econometrics, Springer Dordrecht, 345–381 pp., 10.1007/978- 94-011-2546-8_20, 1992.

Sen, P. K.: Estimates of the Regression Coefficient Based on Kendall's Tau, Journal of the American Statistical Association, 63, 1379-1389, 10.1080/01621459.1968.10480934, 1968.

Shen, L., Jacob, D. J., Zhu, L., Zhang, Q., Zheng, B., Sulprizio, M. P., Li, K., De Smedt, I., González Abad, G., Cao, H., Fu, T. M., and Liao, H.: The 2005–2016 Trends of Formaldehyde Columns Over China Observed by Satellites: Increasing Anthropogenic Emissions of Volatile Organic Compounds and Decreasing Agricultural Fire Emissions, Geophysical Research Letters, 46, 4468-4475, 10.1029/2019gl082172, 2019.

Sillman, S.: The relation between ozone, NOx and hydrocarbons in urban and polluted rural environments, Atmospheric Environment, 33, 1821-1845, 10.1016/s1352-2310(98)00345-8, 1999.

Simon, H., Hogrefe, C., Whitehill, A., Foley, K. M., Liljegren, J., Possiel, N., Wells, B., Henderson, B. H., Valin, L. C., Tonnesen, G., Appel, K. W., and Koplitz, S.: Revisiting day-of-week ozone patterns in an era of evolving US air quality, Atmospheric Chemistry and Physics, 24, 1855-1871, 10.5194/acp-24-1855-2024, 2024.

Wells, K. C., Millet, D. B., Payne, V. H., Deventer, M. J., Bates, K. H., de Gouw, J. A., Graus, M., Warneke, C., Wisthaler, A., and Fuentes, J. D.: Satellite isoprene retrievals constrain emissions and atmospheric oxidation, Nature, 585, 225-233, 10.1038/s41586-020-2664-3, 2020.

Zhang, Y., Wang, Y., Chen, G., Smeltzer, C., Crawford, J., Olson, J., Szykman, J., Weinheimer, A. J., Knapp, D. J., Montzka, D. D., Wisthaler, A., Mikoviny, T., Fried, A., and Diskin, G.: Large vertical gradient of reactive nitrogen oxides in the boundary layer: Modeling analysis of DISCOVER-AQ 2011 observations, Journal of Geophysical Research: Atmospheres, 121, 1922-1934, 10.1002/2015jd024203, 2016.

Zhu, L., Jacob, D. J., Mickley, L. J., Marais, E. A., Cohan, D. S., Yoshida, Y., Duncan, B. N., González Abad, G., and Chance, K. V.: Anthropogenic emissions of highly reactive volatile organic compounds in eastern Texas inferred from oversampling of satellite (OMI) measurements of HCHO columns, Environmental Research Letters, 9, 10.1088/1748-9326/9/11/114004, 2014.